**Historical black carbon deposition in the Canadian High Arctic:**

**A >250-year long ice-core record from Devon Island**

Christian M. Zdanowicz[1], Bernadette C. Proemse[2], Ross Edwards[3,4], Wang Feiteng[5], Chad M. Hogan[2], Christophe Kinnard[6] and David Fisher[7].

[1]Department of Earth Sciences, Uppsala University, Uppsala, 75646, Sweden

[2]School of Biological Sciences, University of Tasmania, Hobart, TAS7001, Australia

[3]Physics and Astronomy, Curtin University, Perth, WA6102, Australia

[4]Depart of Civil and Environmental Engineering, University of Wisconsin, Madison, WI, 53706, USA

[5]Cold and Arid Regions Environment and Engineering Research Institute, Chinese Academy of Sciences, Lanzhou, China

[6]Département des Sciences de l'Environnement, Université du Québec à Trois-Rivières, Trois-Rivières, G9A 5H7, QC, Canada

[7]Department of Earth Sciences, University of Ottawa, 120 University, Ottawa, K1N 6N5, ON, Canada

*Correspondence to*: Christian M. Zdanowicz (christian.zdanowicz@geo.uu.se)

**Abstract.**

Black carbon aerosol (BC) emitted from natural and anthropogenic sources (e.g., wildfires, coal burning) can contribute to magnify climate warming at high latitudes by darkening snow- and ice-covered surfaces, thus lowering their albedo. Modelling the atmospheric transport and deposition of BC to the Arctic is therefore important, and historical archives of BC accumulation in polar ice can help to validate such modelling efforts. Here we present a >250-year ice-core record of refractory BC (rBC) deposition on Devon ice cap, Canada, spanning the years 1735-1992, the first such record ever developed from the Canadian Arctic. The estimated mean deposition flux of rBC on Devon ice cap for 1963-1990 is 0.2 mg m$^{-2}$ a$^{-1}$, which is at the low end of estimates from Greenland ice cores obtained by the same analytical method (~0.1-4 mg m$^{-2}$ a$^{-1}$). The Devon ice cap rBC record also differs from Greenland records in that it shows only a modest increase in rBC deposition during the 20th century, unlike in Greenland where a pronounced rise in rBC occurred from the 1880s to the 1910s, largely attributed to mid-latitude coal burning emissions. The deposition of contaminants such as sulfate and Pb increased on Devon ice cap in the 20th century but no concomitant rise in rBC is recorded in the ice. Part of the difference with Greenland could be due to local factors such as melt-freeze cycles on Devon ice cap that may limit the detection sensitivity of rBC analyses in melt-impacted core samples, and wind scouring of winter snow at the coring site. Air back-trajectory analyses also suggest that Devon ice cap receives BC from more distant North American and Eurasian sources than Greenland, and aerosol mixing and removal during long-range transport over the Arctic Ocean likely masks some of the specific BC source-receptor relationships. Findings from this study suggest that there could be a large variability in BC aerosol deposition across the Arctic region arising from different transport patterns. This variability needs to be accounted for when estimating the large-scale albedo lowering effect of BC deposition on Arctic snow/ice.

**1 Introduction**

The deposition of light-absorbing carbonaceous particles emitted by the incomplete combustion of biomass and fossil fuel can decrease the albedo of Arctic snow- and ice-covered surfaces, thereby amplifying high-latitude warming driven by the buildup of greenhouse gas emissions (AMAP, 2011; Bond et al., 2013). The widely used expression "black carbon" (BC) designates the insoluble,

refractory fraction of these aerosols that is largely made of graphic elemental carbon and strongly absorbs light at visible to near-infrared wavelengths (Petzold et al., 2013). Along with sulfate ($SO_4^{2-}$), BC is one of the main short-lived climate pollutants being targeted for mitigation and control under multinational legal agreements (Quinn et al., 2008; AMAP, 2015).

In order to evaluate how past and future BC emissions have affected, and will affect, climate forcing in the Arctic, global atmospheric climate models can be used to simulate the transport and deposition of BC aerosols in this region (Koch et al., 2011; Skeie et al., 2011; Lee et al., 2013; Jiao and Flanner, 2016). At present, simulated BC dispersion suffers from large biases, either positive or negative, compared with observational data on BC in Arctic air and snow (Jiao et al., 2014). Validating model simulations is difficult because of the scarcity of such observations across the Arctic. Direct monitoring of atmospheric BC is so far limited to a few decades and at a few stations (Hirdman et al., 2010; Gong et al., 2010), and geographic surveys of BC in snow and ice are rare and difficult to conduct over the vast Arctic region (e.g., Doherty et al., 2010).

Ice cores drilled from the accumulation area of glaciers and ice caps can be used as surrogates for direct atmospheric observations, as they contain archives of BC and other aerosol species deposited in snow over many centuries (McConnell, 2010). At present, ice-core records of BC deposition in the Arctic region are only available from Greenland (McConnell et al., 2007, McConnell and Edwards, 2008; Zennaro et al., 2014; Sigl et al., 2015) and from Svalbard (Ruppel et al., 2014). Here, for the first time, we present a historical record of BC deposition in the Canadian Arctic, developed from a core drilled on Devon Island ice cap, and spanning the years ~1735-1992. The Devon ice cap BC record presents some striking differences from Greenland ice-core records of rBC developed by the same methods. We discuss the possible reasons for these differences, and consider the implications with respect to regional BC transport and deposition patterns in the Arctic region.

**2 Study site**

At latitude 75° N, Devon ice cap (14,400 km$^2$) occupies a central position in the eastern Canadian Arctic Archipelago and lies 275 km from the Greenland coast across northern Baffin Bay. The ice cap has been studied for half a century (Boon et al., 2010) and was previously drilled to obtain records of climate and atmospheric contaminants (e.g., Goto-Azuma and Koerner 2001; Shotyk et al., 2005; Kinnard et al., 2006). However, no record of BC deposition was ever developed from

this or any other site in the Canadian Arctic. The core used in the present study (DV99.1) was
obtained in April 1999 by the Geological Survey of Canada (GSC) at the top of a dome (75.32° N,
81.64° W, 1903 m.a.s.l.) located 25 km to the east of the ice cap's main dome and true summit
(~1930 m.a.s.l) (**Fig. 1**). The coring site lies above the present-day equilibrium line which, based
on long-term mass balance observations, has a mean altitude of 1150 m a.s.l.. The mean annual air
temperature at the summit of Devon ice cap is -22 °C (Bezeau et al., 2013), and the estimated mean
accumulation rate ($\dot{A}$) at the DV99.1 coring site is 0.14 m ice a$^{-1}$, or 0.16 m H$_2$O a$^{-1}$ (see below).

**3 Materials and methods**

**3.1 Core sampling and analyses**

The DV99.1 core was recovered in 0.4 to 1.1-m long increments (average 0.9 m), with a diameter
of 9.8 cm. The uppermost 2.8 m of the core were made of crumbly firn, and could not be preserved.
At the time of coring, the solid-state DC electrical conductivity (EC) of the core was measured
continuously using a hand-held system with parallel electrodes, as described in Zheng et al. (1998).
The EC profiling started at a depth of 12.38 m, because section of cores above this were of brittle
firn that provided inadequate electrode contact for the hand-held instrument. The core was shipped
and stored in freezers at the GSC ice-core laboratory in Ottawa. There, it was sampled at 5- to 20-
cm resolution for the determination of stable oxygen isotope ratios ($\delta^{18}$O) by mass spectrometry at
the University of Copenhagen. Later, 57 discrete sub-samples from depths below 29 m were
analyzed for lead (Pb) and other trace metals, as reported in Zheng et al. (2007). The remaining
cores were stored frozen (-20ºC) inside sealed polyethylene bags, until archived core segments
between 2.8 and 48 m depths were selected for this study and shipped, still frozen, to Curtin
University in Australia for BC analyses. These combined core segments were estimated to span
>250 years, as explained below.
Sample preparation and analysis was conducted between 6 and 11 Dec. 2012 at the Trace
Research Clean Environmental facility at Curtin University. The facility consists of a large class
space containing multiple class 10 laboratory modules including a -20ºC walk-in freezer within
a general lab space (also class 10). The space was specifically designed for trace metal and particle
work on ice cores (e.g., Burn-Nunes et al. 2011, Ellis et al., 2015, 2016; Tuohy et al., 2015;
Vallelonga et al., 2017 ). The DV99.1 core sections were cut into sub-samples with a ~2.5 × 2.5
cm cross-section, which were processed in an ice-core melter coupled to a Continuous Flow

Analysis (CFA) system (see supplement, **Fig. S1**). Ice core preparation was carried out in the walk-in freezer, while processing in the CFA system was conducted in the general lab class 100 space. The CFA melter system was similar to that described by McConnell et al. (2002) with the exception that the ice core melter head was made from aluminum. The method used to quantify BC in the ice core was essentially the same as used by others for the analysis of Greenland and Antarctic cores (Bisiaux et al., 2012; McConnell et al., 2007, McConnell and Edwards, 2008; Zennaro et al., 2014). Meltwater from the CFA system was aerosolized and desolvated with a U5000AT ultrasonic nebulizer (CETAC Technologies, Omaha, NE, USA) and injected into a single-particle intracavity laser-induced incandescence photometer (Schwarz et al. 2010; SP2, Droplet Measurement Technologies, Boulder, CO), which measured the mass concentration of BC particles in the meltwater flow. Instrumental settings are given in the supplement (**Table S1**). Following Petzold et al. (2013), we refer to the BC fraction measured by this method as *refractory BC* (rBC), reported here in mass concentration units of ng $g^{-1}$.

On each day of analysis, a log journal was created. Every piece of the DV99.1 core was carefully measured in its length prior to analysis. During CFA, the time of each break between two ice core pieces was recorded, making it possible to reconcile the rBC record of each piece based on the time-depth log. The flow rate of the CFA to the nebulizer was controlled by oversupplying a <1 mL debubbling vessel with excess water, allowing the instrument to maintain a very constant flow rate. External calibration of the SP2 nebulizer system was achieved using eight standards of 100% black carbon ink (Ebony MIS, EB6-4 K; **Fig. S2**) spanning a concentration range of 0 to 20 ng $g^{-1}$. The standards were analyzed each day before and after ice core analysis and the results were compared to assess the stability, reproducibility, and measurement uncertainty of the SP2. Additional details and calibration curves (**Fig. S3**) are provided in the supplement, and potemtial sources of uncertainties in the results are discussed under section 3.3 below.

To compare the DV99.1 record of rBC with that of other aerosol species, we used glaciochemical data obtained from two other cores drilled from the summit area of Devon ice cap in 1998 (core DV98.3) and 2000 (core DV2000) (**Fig. 1; Table 1**). The DV98.3 core was sampled continuously and analyzed for eight major ionic species by ion chromatography, as described in Kinnard et al. (2006). In this study, we used $SO_4^{2-}$, sodium ($Na^+$), calcium ($Ca^{2+}$), potassium ($K^+$) and ammonium ($NH_4^+$) data obtained from the top 85 m of the core, which had been sampled at 3- to 12-cm resolution. The non-sea salt fraction of sulfur (nssS) was estimated from $Na^{2+}$ using the

mean surface seawater composition of Pilson (2012), and the biomass burning fraction (BB) of $K^+$
was estimated from $Na^{2+}$ and $Ca^{2+}$ as: $[K^+]_{BB} = [K^+] - (0.038 \times [Na^+]) - (0.04 \times [Ca^{2+}])$, following
Legrand et al. (2016). The DV2000 core was drilled at the same site as the DV98.3 core, and was
analyzed for lead (Pb) and other metals, as reported in Shotyk et al. (2005). The remaining archived
volume from cores DV98.3 and DV2000 was, however, insufficient to carry out rBC analyses,
which is why core DV99.1 was used for this purpose.

## 3.2 Age models

Annual layers are not easily resolved in cores from Canadian Arctic ice caps, partly owing to
relatively low $\dot{A}$, but also to the effects of wind and/or summer surface melt. Therefore, age models
developed for these cores are commonly based on a variety of alternative methods. For the DV98.3
and DV99.1 cores, an ice-flow model (Dansgaard and Johnsen, 1969) was used, constrained by the
total ice thickness obtained from ice-radar measurements or from borehole depths, and by the
estimated $\dot{A}$ at each coring site. For the DV98.3 core, the age model was further constrained by
approximate layer counting using $\delta^{18}O$ and glaciochemical data at shallow depths, and, at greater
depths, using reference horizons from bomb radioactive fallout (1963; Pinglot et al. 2003) and from
historical volcanic eruptions, including that of Laki, Iceland, in 1783 [All given dates are C.E.],
which is one of most recognizable historical volcanic signals recorded in EC and/or $SO_4^{2-}$ records
of other Canadian Arctic ice caps (e.g., Zheng et al., 1998; Goto-Azuma et al., 2002). The age
model in the upper 48 m of the DV99.1 core was constrained using a reference horizon provided
by a large EC (acidity) spike at a depth of 42.60 m (29.56 m ice equivalent), which was attributed
to the 1783 Laki eruption (**Fig. 2**). This model gives an estimated maximum age of 1735 for the
section of the DV99.1 core used in the present study, and the last year in the record is 1992. The
age model also gave an acceptable agreement between profiles of various measured parameters in
the DV98.3 and DV99.1 cores (**Fig. S5-S6**). The DV2000 core was drilled at the same site as the
DV98.3 core and used the same age model. The two cores were correlated using measurements in
the DV2000 that allowed identification of the 1958 (16.5 m depth) and 1963 (13.5 m depth)
radioactive layers (Krachler et al., 2005). The DV2000 core was estimated to extend back to 1842.
Using the Laki 1783 reference layer, the estimated $\dot{A}$ at the DV99.1 site is 0.14 m ice a$^{-1}$ (0.16
m $H_2O$ a$^{-1}$) which is lower than at the ice cap summit (~0.25-0.28 m $H_2O$ a$^{-1}$) or at sites elsewhere
in the accumulation zone (0.17-0.25 m $H_2O$ a$^{-1}$; Colgan and Sharp, 2008). The most likely
explanation is partial scouring of winter snow layers by downslope winds at the DV99.1 site, as
also observed on parts of Agassiz ice cap (Fisher et al., 1983). This is supported by a comparison
of the $\delta^{18}O$ measurements in the DV99.1 and DV98.3 cores, which shows that $\delta^{18}O$ variations in
the DV99.1 core are truncated of their most negative ("coldest") values relative to the DV98.3 core
(**Fig. S7**). An estimate of the amount of snow lost by wind scouring at the DV99.1 site can be made
from the difference in the amplitude of the $\delta^{18}O$ data at the DV98.3 and DV99.1 sites, and from $\dot{A}$
at the DV98.3 site, following Fisher and Koerner (1988). The calculation suggests that ~40-45 %
of the annual snow accumulation is removed by wind at this site, compared to the summit of Devon
ice cap.
**3.3 Quantifying uncertainties in the rBC record**
Analyses of rBC in the DV99.1 core were performed at high depth resolution, producing ~55-80
data points per meter over most of the core's length. The data were subsequently averaged over
discrete depth increments equivalent to ~1- and ~10-year intervals, respectively, based on the core's
age model. In this paper, annually-averaged figures are used for illustrative purposes only, as
individual years can not be confidently resolved in the DV99.1 core. Down-core variations of rBC
in the ice core are the result of a combination of processes, including temporal changes in
atmospheric deposition rates (fluxes, abbreviated $F$), spatial variations of deposition of aerosols in
snow, and post-depositional modifications (e.g., by wind scouring or summer surface melt).
Additional uncertainties in the rBC data come from the age model of the ice core (**Fig. 2**), and from
limitations of the analytical method.

The largest uncertainty with regards to the rBC analysis is due to the nebulization /

desolvation step before the SP2 analysis. At the time of this study we had adopted nebulizer /
desolvation systems used as a front end to inductively-coupled plasma mass spectrometers (ICP-
MS). These systems are designed to deliver appropriate aerosol size distributions for analysis in
the ICP-MS. Schwarz et al. (2012) and Wendl et al. (2014) report rBC size-dependent losses during
nebulization / desolvation for several types of nebulizer desolvation systems. The study found that
the system used in this investigation has a poor transport efficiency for rBC particles with a volume
equivalent diameter >500 nm. Hence rBC data from the DV99.1 core should be considered with
this limitation (see section 4.2 for a discussion). Other published ice core data sets from Greenland
(for example Mc Connell et al., 2007) also suffer from this limitation, but are at least comparable.
Further research is required to assess the true size distribution of rBC deposition to the Devon ice
cap and other Arctic sites.

Uncertainties in the DV99.1 age model are primarily due to the potential dating error of the
Laki 1783 layer in the EC profile, and to interannual variations in the net accumulation rate at the
ice-coring site. The relationship between true depth and ice-equivalent depth is nearly linear in the
DV99.1 core down to 48 m, which suggests a steady firn densification rate over the corresponding
time interval, with no signs of dynamically-induced changes in the vertical strain rate. For the 1783
layer, we assumed a possible dating error of $\pm 5$ years, corresponding to a registration error of $\sim\pm 1$
m at the 42.6 m EC peak. The interannual variability in the accumulation rate was estimated from
an array of shallow cores (Colgan and Sharp, 2008) and from winter mass balance measurements
since 1961 (data available through the World Glacier Monitoring Service). This information was
used in a Monte Carlo simulation in Matlab™ with 1000 realizations to compute confidence limits
(CL) on the decadally-averaged rBC data. Briefly, a constrained random walk algorithm was used
to estimate the probabilistic distribution of the true age at any depth in the core from the surface
down to the Laki 1783 layer (Kinnard et al., 2006). Interannual variations in $\dot{A}$ were considered to
behave as a stationary, autoregressive blue noise process with a lag-one serial autocorrelation
coefficient of -0.5 to -0.3, based on empirical data presented by Fisher et al. (1985). A population
of 1000 alternative age models was thus generated. From each of these, 10-year averages of the
rBC data were computed, and 95 % CL were calculated for the geometric mean rBC concentration
in each decade (**Fig. S8**). Expressed as a coefficient of variation (CV), the estimated uncertainty
on the decadal rBC averages that arise from age model errors varies from 3 to 23 % (median 6 %),
depending on the decade considered.

The spatial variability of BC deposition on Canadian Arctic ice caps is unknown. An estimate
for Devon ice cap can be made from major ion analyses on shallow cores (Colgan and Sharp, 2008;
**Fig. 1**). In these cores, the spatial coefficient of variation (CV) on the annual $SO_4^{2-}$ deposition
averages 42 % (range 17-100 %) over a period of $\sim$40 years. Here, we make the assumption that
deposition of rBC on Devon ice cap shares the same spatial variability as $SO_4^{2-}$, an aerosol species
which, like BC but unlike others such as nitrate ($NO_3^-$), is not subject to re-emission from snow to
air. While the spatial variability may be large on an annual basis, Monte Carlo simulations results
show that averaging the rBC data over 10-year intervals reduces its effect on the geometric mean
rBC uncertainty to a few % (CV) in any decade (**Fig. S8**). The potential impact of post-depositional
modifications in the rBC record is discussed under section 4.2 below.

## 4 Results and discussion

### 4.1 The DV99.1 record of rBC

The depth profile of rBC measured in the DV99.1 core is shown in **Fig. 3**. The probability
distribution of rBC concentrations is approximately log-normal (**Fig. S9**), and we therefore use
both the arithmetic and geometric means ($\mu$, $\mu_g$), as descriptive metrics for these data. Over the
entire core length, rBC concentrations average $1.8 \pm 3.9$ ng g$^{-1}$ ($\mu_g = 0.8$ ng g$^{-1}$) with a maximum
of 74.0 ng g$^{-1}$. The mean rBC concentration is approximately constant between 42 and 15 m depths,
and decreases gradually at shallower depths to reach ~1.0 ng g$^{-1}$ ($\mu_g = 0.5$ ng g$^{-1}$) in the uppermost
meter of core. Concentrations below 42 m show a comparatively larger variability and a greater
range of values (**Fig. S10**).
In Greenland cores, rBC deposition rose in the 1880s, peaked in the 1910s-20s, and decreased
thereafter (McConnell et al., 2007), in step with historical changes in coal-burning BC emissions
from North America and Europe (Novakov et al., 2003; Bond et al., 2007; Lamarque et al., 2010).
In south-central Greenland, the early 20th century rise in rBC and nssS was also accompanied by
increased deposition of Pb and other trace metals (McConnell and Edwards, 2008). Measurements
from the DV98.3 and DV2000 ice cores (**Fig. 4**) show that Devon ice cap also experienced
increased atmospheric deposition of $SO_4^{2-}$ and Pb and during the 20$^{th}$ century, peaking between the
1960s and 1980s, and followed by a decline, consistent with trends in mid-latitude anthropogenic
emissions from fossil fuel combustion. However, unlike in Greenland, the DV99.1 core shows no
large, sustained increase in rBC concentration concomitant with that of $SO_4^{2-}$ or Pb. There is a
modest rise in mean rBC concentrations from the early 1800s to the mid-20th century, but it is
much more gradual and of lesser magnitude than the rBC rise observed in ice-core records from
Greenland, although the relative timing and magnitude of these increases differ between core sites
(**Fig. 5** and **6**). In the DV99.1 ice core, the highest mean rBC concentrations for the 20th century
occur in the decade 1960-70 ($\mu = 4.7$ ng g$^{-1}$, $\mu_g = 1.7$ ng g$^{-1}$), but these are not unprecedented, and
comparable mean concentrations occur in the earliest part of the record, in the decade 1780-1990
(**Fig. S10**).
The DV99.1 rBC record also shows a pronounced decline in rBC concentration in the late
20th century, but it occurs after the 1960s, which is later than in most Greenland cores, except at
Humboldt (**Fig. 5** and **6**). This difference in timing could, however, be due to uncertainties in the
DV99.1 chronology compared to that of well-dated Greenland cores. The DV99.1 mean rBC
concentrations over the period 1960-1990 ($\mu = 0.6\text{-}1.0$ ng g$^{-1}$; $\mu_g = 0.3\text{-}0.5$ ng g$^{-1}$) are lower than
in the early modern industrial period, in the early 19th century ($\mu = 1.0\text{-}3.0$ ng g$^{-1}$; $\mu_g = 0.7\text{-}1.6$ ng
g$^{-1}$). The only Greenland ice core in which a similar situation occurs is from the ACT2 site (66°N,
**Fig. 5**). Neither winter mass balance measurements, nor reconstructed interannual changes in $\dot{A}$ on
Devon ice cap (Colgan and Sharp, 2008) show any sustained long-term trend since the early 1960s,
and the decrease in rBC concentration in the DV99.1 core during this period can therefore not be
ascribed to changing precipitation rates on the ice cap. It seems more likely that the decrease is at
least in part due to a declining burden of atmospheric BC in the Canadian High Arctic since the
1960s (Gong et al., 2010). However, there are several methodological, site-specific and regional-
scale factors that must be taken into account when interpreting the DV99.1 rBC record. These are
discussed below.
**4.2. Methodological and site-specific factors**
Observations of atmospheric BC at Alert on Ellesmere Island (82° N, **Fig. 1**) show a seasonal cycle
with airborne concentrations peaking during winter and spring months (December-March) and
declining to their minimum in summer and early autumn months (June-September) (Gong et al.,
2010). Most BC deposition in snow is thought to occur in spring and summer, when increased
cloudiness promotes in-cloud scavenging and wet deposition of the hydrophilic fraction of BC
(Garrett et al., 2011; Browse et al., 2012; Shen et al., 2017). In the interior of the Greenland ice
sheet, the seasonal cycle of BC deposition is well-preserved in snow and firn layers (e.g.,
McConnell et al., 2007). This is not the case at the DV99.1 core site on Devon Island. Even in the
uppermost part of the core, where some seasonal $\delta^{18}$O variations can be detected, there is no
recognizable seasonal pattern of rBC concentration peaks (**Fig. S11**). This is likely the result of the
combined effects of wind scouring/mixing of surface snow (as described earlier) and of summer
surface melt. The question therefore arises whether such processes could also have obliterated or
masked a 20$^{th}$ century anthropogenic signal in the DV99.1 rBC record.

The seasonally-resolved ice core record from site D4 in Greenland (71°N; **Fig. 5**) shows that during the historical period of enhanced anthropogenic BC pollution in the Arctic, from the late 19th to mid 20th centuries, rBC deposition increased in both summer and winter (McConnell et al., 2007). If the Canadian High Arctic was impacted by airborne BC pollution in a similar way, one would expect to find a marked increase in rBC concentrations in the DV99.1 core during the early 20th century, even if winter snow layers were scoured away by wind. To verify this, we performed a simple simulation in which we generated synthetic time series of rBC deposition spanning the period 1800-1990, with a seasonal cycle superimposed on baseline inter-decadal variations similar to those observed in the Greenland D4 ice-core record. Winter rBC deposition peaks in the series were represented using a log-Gaussian function, and their amplitude was allowed to vary from year to year to produce a range of temporal variations comparable to, or lower than, that seen in the Greenland D4 core. Winter deposition peaks were then randomly truncated by 30-60 % (mean 45 %) to simulate the effects of wind scouring on the record, and 5-year running means were computed from the resulting data, the smoothing being used to simulate the effects of post-depositional snow layer mixing by wind. Results of these experiments show that even if the wintertime rBC deposition peaks between November to May were largely truncated by wind, the low-frequency baseline variation would still persist, and should be recognizable above the remaining interannual signal variance (**Fig. 7**). It therefore seems unlikely that wind scouring alone would completely obliterate this rBC signal in the DV99.1 record, not unless the amplitude of the seasonal cycle of atmospheric BC deposition on Devon ice cap is much lower than observed at Alert or in Greenland (Gong et al., 2010; Massling et al., 2015).

Unlike much of central Greenland, the summit of Devon ice cap is subject to partial melting at the surface during summer months, and meltwater can percolate and refreeze into the underlying snow and firn to form infiltration ice features ("melt layers"). The volumetric percentage of melt layers in core DV99.1 was measured by Fisher et al. (2012) as a proxy for past summer warmth. These data show that surface melt rates at the coring site increased abruptly in the mid-19th century following the end of the Little Ice Age cold interval, and have since averaged 22 % (median 19 %), occasionally exceeding 50 % in the 20th century (**Fig. 4**). The DV99.1 coring site is above the present-day upper limit of the superimposed zone (~1400 m a.s.l.; Gascon et al., 2013) and the firn there is >60 m thick, so it is very unlikely that there is any net loss by runoff at this location: any meltwater produced in the summer must refreeze in the firn. However, even without net losses, one

must consider whether meltwater percolation and refreezing could account for the limited variability in the DV99.1 rBC record during the 19th and 20th centuries.

The post-depositional mobility of BC particles in melting snow is not well known, and likely depends on the hydrophobicity of these particles, which is largely influenced by the presence or absence of surface coatings, for e.g., with $SO_4^{2-}$ (Liu et al., 2011, 2013). Doherty et al. (2013) investigated the vertical redistribution of BC and other light-absorbing particles in snow and firn near Dye 2 (66° N; ~2100 m a.s.l.; **Fig. 5**) in a part of the Greenland ice sheet's percolation zone where melt layers >10 cm thick are now commonly found (de la Peña et al., 2015; Machguth et al., 2016). Only very limited vertical redistribution of BC was observed in the snow and firn, and surface melt and percolation did not obliterate seasonal variations of BC in the firn stratigraphy. Doherty et al. (2013) attributed this result to the low scavenging efficiency of these particles by meltwater (~20-30 %). At the DV99.1 site on Devon Island, ice layers >10 cm are comparatively very rare, but $\dot{A}$ (0.14 m a$^{-1}$) is only half of that in the Dye 2 area (~0.32 m a$^{-1}$; Buchardt et al., 2012). Therefore surface melt could mask some seasonal variations of rBC in the firn.

The depth at which meltwater could percolate in firn at the DV99.1 site is not known precisely over the time period covered in the rBC record. The thickness of the firn zone there (>60 m) is much greater than at Lomonosovfonna summit, Svalbard, for example (~25 m; Kekonen et al., 2005). If we accept the estimated depth range of 0.5-2 m for meltwater-induced relocation of water-soluble ions at Lomonosovfonna summit for 2000-07 reported by Vega et al. (2016), then it is highly unlikely than relocation of rBC particles could be deeper at the DV99.1 site. The summit of Devon ice cap is ~650 m higher than the Lomonosovfonna summit (1250 m a.s.l.), has a much lower mean annual surface temperature (-22°C, compared to ~-10 to -12 °C at Lomonosovfonna; W. van Pelt, pers. comm.), and the 10-m firn temperatures on Devon ice cap summit were, in 2012, < -15 °C (Bezeau et al., 2013), while those at Lomonosovfonna were within -2 to -3° of zero C in 1997 (van de Waal et al., 2002). Attempts were also made to quantify post-depositional deposition of ions and/or particles by melt/percolation on Penny ice cap on Baffin Island (66° N; Grumet et al., 1998; Zdanowicz *et al.*, 1998), where estimated summer melt rates over the last 150 years are much higher (40-100 %) than at the DV99.1 site (Zdanowicz et al., 2012). On Penny ice cap during the mid-1990s, ions and particles were estimated to be redistributed over depths of 3-5 m. A plausible, conservative estimate of the maximum melt-induced relocation depth at the DV99.1 site for the time period of interest might therefore be 3 m (firn depth). With a mean accumulation rate

of 0.16 m $H_2O$ $a^{-1}$ at the site, soluble impurities could be offset by meltwater percolation in the core
by 5-8 years relative to their true depositional depth/age, and probably less for rBC particles given
their hydrophobicity. In this paper, we focus on inter-decadal variations in rBC concentrations. At
such a time-averaging window length, the effect of impurity relocation by melt should largely even
out.

There is, however, another consideration. Unlike in the Doherty et al. (2013) study, rBC

concentrations in the DV99.1 core were measured by SP2, and the detection efficiency of this
method for BC in liquid samples depends on the type of nebulizer used for inflow. As previously
mentioned, Schwarz et al. (2012) and Wendl et al. (2014) have shown that the relative
areosolization efficiency of rBC by the U5000AT ultrasonic nebulizer used in the analysis of the
DV99.1 core drops rapidly for particles with a volume-equivalent diameter >500 nm (~ 10%
efficiency at a volume-equivalent diameter of 600 nm). Coagulation and agglomeration is known
to increase the size of BC particles during thaw and refreezing of snow (Schwarz et al., 2013), and
this raises the possibility that the SP2 may underestimate the true mass concentration of BC
particles in those parts of the DV99.1 that contain icy layers (**Fig. 3**). Some of the central and
northern Greenland sites (e.g., Summit, NEEM) from which ice-core rBC records were developed
by the SP2 method (**Fig. 5**) certainly experience less or no summer surface melt, compared to
Devon ice cap, and rBC particles in firn at these sites are probably largely unaffected by post-
depositional coagulation. Other coring sites located in southern Greenland (ACT2, D4) or at lower
elevations (Humboldt) may experience some surface melt and refreezing in summer, but statistics
on ice layer frequency at these sites are unpublished so this cannot be verified. While melt-
refreezing may have contributed to mask some historical variations in atmospheric BC deposition
at the DV99.1 site, it is uncertain if this alone can account for the low rBC concentrations in the
DV99.1 core, when compared to Greenland records analyzed using the same methods. One
conflicting observation is that the lowest rBC concentrations in the core are found in the uppermost
5 m, and this part of the core actually contains fewer ice layers than deeper sections (between 7-32
m) in which some of the highest rBC concentrations are measured (**Fig. 3**).
**4.3 Regional-scale factors**
Other reasons for the differences between the DV99.1 and Greenland rBC records (**Fig. 5** and **6**)
may be found in the atmospheric transport paths that deliver BC to the Canadian High Arctic,

relative to Greenland. Shindell et al. (2008) used multiple atmospheric transport models to investigate the sensitivity of near-surface airborne BC concentrations in the Arctic to regional anthropogenic emissions. They found that Europe and North America likely contribute equally to BC deposition over Greenland, whereas the central and Russian sectors of the Arctic are more impacted by European emissions. Atmospheric BC in the Canadian High Arctic may be affected by both European and North American emissions, but the region is expected to be less sensitive to changes in these emissions compared to other parts of the Arctic, partly because it is very remote from all BC source regions (Shindell et al., 2008; their Fig. 9 and 10).

Sharma et al. (2006) and Huang et al. (2010) used air back-trajectory analyses to investigate the probable source regions of BC detected at Alert in winter and spring, and identified Russia and Europe as dominant, followed by North America. The summit of Devon ice cap is 1000 km further south and ~1.9 km higher, and could thus be affected by a different mix of BC source contributions than Alert. To verify this, and also to contrast the situations of Devon ice cap and Greenland, we computed ensemble 10-day air back-trajectories from both Devon ice cap summit and from Summit, Greenland, using the HYbrid Single-Particle Lagrangian Integrated Trajectory model (HYSPLIT v.4) of the NOAA Air Resources Laboratory (Draxler and Hess, 2014, Stein et al., 2015). As input, we used meteorological fields of the NCEP-NCAR 50-year reanalysis product, which are available on a global 2.5 × 2.5° grid at 6-hourly temporal resolution (Kistler et al., 2001). Back-trajectories starting daily at 12:00 PM UTC were computed over the period 1948-1999. Unlike Sharma et al. (2006) and Huang et al. (2010), however, we did not use trajectory clustering, because results are highly sensitive to the quality and density of meteorological data coverage used in trajectory computations, and to the arrival height of trajectories (i.e., starting point of back-trajectories; Kassomenos et al., 2010; Su et al., 2015). Instead, we computed probability density maps or air parcel residence time from all combined trajectories over an equal area grid with 200 × 200 km resolution, following a methodology analog to that of Miller et al. (2002).

Results (**Fig. 8**) show that for 10-day transport periods, air parcels arriving at Greenland Summit are more commonly advected from the south-southwest than from other directions, and frequently reach central Greenland after transiting over the North Atlantic, consistent with earlier findings by McConnell et al. (2007; Their Fig. S1). In contrast, air that reaches the summit of Devon ice cap comes more frequently from the west-northwest, and transits over the Arctic Ocean, which agrees with findings from analyses of low-level air transport to Devon ice cap by Colgan

and Sharp (2008) for the period 1979-2003. It is therefore likely that a large part of BC transported
to Devon ice cap is from regional emission sources located in northwestern North America and/or
in the central or eastern parts of Eurasia.

Smoke plumes from forest or grassland fires, natural or provoked, can reach the Arctic and

contribute to BC pollution, particularly during summer (Stohl et al., 2006; Paris et al., 2009;
Warnecke et al., 2009; Quennehen et al., 2012; Zennaro et al., 2014; Hall and Loboda, 2017). Back-
trajectory analyses of BB aerosols detected at Eureka on Ellesmere Island (80° N; **Fig. 1**) indicate,
unsurprisingly, that boreal forest/grassland regions of Russia and Canada are the dominant source
regions for these long-range plume transport events, followed by north-central USA and Alaska
(Viatte et al., 2015). To investigate the impact of forest/grassland fire emissions on BC deposition
to Devon ice cap, we compared the DV99.1 rBC record with reconstructed variations in fire
frequency and/or burned area across Canada and Russia during the 19th and/or 20th centuries (**Fig.**
**9**; data from Girardin, 2007; Girardin and Sauchyn, 2008; Girardin et al., 2006; and Mouillot and
Field, 2005). On an inter-decadal time scale, no statistically meaningful correlations ($p < 0.05$)
could be identified between the DV99.1 rBC record and the fire histories. If wildfire emissions
contribute to rBC deposition on Devon ice cap, these contributions are either too small and/or
mixed in the DV99.1 record to be correlated with variations in fire frequency or burned area in the
source regions.

Aerosol species such as $K^+$ or $NH_4^+$ are commonly associated with BB emissions, and are

often used as BB tracers in polar snow (Simoneit, 2002; Legrand et al., 2016). Cheng (2014)
identified sectors of south-central Russia and Kazakhstan as source regions for both BC and $K^+$
aerosols transported to Alert between 2000 and 2002. However, we did not find any significant
correlations ($p < 0.05$) between inter-decadal variations of rBC in the DV99.1 core and either
$(K^+)_{BB}$ or $NH_4^+$ in the DV98.3 record (**Fig. S12**). Whatever contributions BB emissions make to
$(K^+)_{BB}$ or $NH_4^+$ deposition on Devon ice cap, these do not covary directly with rBC deposition,
possibly due to different post-depositional relocation of these impurities in the DV98.3 and DV99.1
cores, but also to mixing from multiple emission sources. For example, ammonia ($NH_3$) emissions
from seabird colonies near Baffin Bay may be a larger regional source of $NH_4^+$ to Devon ice cap
than distant wildfires (Wentworth et al., 2016).

### 4.4 Atmospheric BC deposition rates

In 90 % of the analyzed DV99.1 core, rBC concentrations are < 3 ng g$^{-1}$, and in the uppermost section of the core (depths 3-4 m), they are mostly ≤1 ng g$^{-1}$. These concentrations are very low compared with the 8-14 ng g$^{-1}$ reported by Doherty et al. (2010) for seasonal snow sampled across the Canadian Arctic in 2009. Part of the apparent discrepancy may be due to differences in analytical methods: The BC concentrations in snow reported by Doherty et al. (2010) were measured using a spectrophotometric technique which tends to yield larger mass concentrations relative to the SP2 method (Schwarz et al., 2012). Also, as stated earlier, rBC levels measured in the DV99.1 core may underestimate actual deposition due to wind scouring of winter snow. Atmospheric BC deposition over the summit region of Devon ice cap could also be lower than near sea level, where most of Doherty et al.'s (2010) samples were obtained, because most of the ice cap's accumulation area (≥ ~1150 m a.s.l.) is above the typical altitude range of low-level Arctic stratocumulus cloud decks which promote aerosol scavenging (Browse et al., 2012).

Despite the aforementioned uncertainties, we estimated the average late 20th century atmospheric flux of rBC ($F_{rBC}$) over the summit region of Devon ice cap using measurements of rBC concentrations in the DV99.1 core for 1963-1990, and data on spatial and temporal variations of $\dot{A}$ from Colgan and Sharp (2008) and from winter mass balance surveys carried out over the ice cap since the early 1960s. The period 1963-1990 was selected because the 1963 radioactive layer in Devon ice cap firn provides a reference level to constrain estimates of $\dot{A}$ (Colgan and Sharp, 2008). Our calculations yield a mean $F_{rBC}$ of 0.2 ± 0.1 mg m$^{-2}$ a$^{-1}$. If $\mu_g$, rather than $\mu$, is used to estimate average rBC concentrations, the estimated $F_{rBC}$ is only slightly lower (0.1 mg m$^{-2}$ a$^{-1}$). And if the measured concentrations of rBC are underestimated by 20-40 % due to selective wind scouring of winter snow layers and/or to inadequate detection by the SP2 instrument, the adjusted figures for $F_{rBC}$ are only slightly higher, ranging from 0.2 to 0.3 mg m$^{-2}$ a$^{-1}$.

These estimates are at the low end of calculated $F_{rBC}$ in Greenland cores over the same period, which vary from ~0.1 to ~4 mg m$^{-2}$ a$^{-1}$ (Lee et al., 2013). They are also much lower than historical fluxes of elemental carbon in the Norwegian Arctic inferred from a Svalbard ice core, which range from ~3 to nearly 40 mg m$^{-2}$ a$^{-1}$ between 1960 and 2004 (Ruppel et al., 2014; see **Fig. 5** for location). Overall, rBC or elemental carbon concentrations in Arctic firn and ice cores increase with mean $\dot{A}$ (**Fig. S13**), likely reflecting the amount of precipitation scavenging in different geographic sectors

(Garrett et al., 2011; Browse et al., 2012). Hence the low $F_{rBC}$ at the DV99.1 site may be partly due
to the low $\dot{A}$ on Devon ice cap (0.14-0.25 m a$^{-1}$; Colgan and Sharp, 2008), notwithstanding the
effect of other factors.
**5 Summary and conclusions**
We developed a >250-year time series of atmospheric rBC deposition from Devon ice cap spanning
the years ~1735-1992. The rBC ice core record (core DV99.1) is the first from the Canadian Arctic,
and it supplements existing ice-core records of rBC or elemental carbon from Greenland and
Svalbard. The DV99.1 record differs from Greenland records developed by the same analytical
methods in that it only shows a very modest and gradual rise in rBC deposition through the 19th
and early 20th century, unlike most Greenland ice cores, in which there is large, well-defined rise
in the 1880-90s, peaking in the 1910s. This rise was attributed to BC emissions from coal
combustion, which also emitted $SO_2$ and trace metals such as Pb (McConnell et al., 2007). Ice cores
from Devon ice cap (DV98.3, DV2000) show that the deposition of $SO_4^{2-}$ and Pb also increased
there during the 20th century, but the DV99.1 core shows no concomitant rise in rBC.

We suggest that differences between the DV99.1 and Greenland rBC records are due to a

combination of methodological, site-specific and regional-scale factors. The site DV99.1 coring
site is subject to summer melt-freeze cycles, and this may lead to underestimation of true rBC
concentrations by the SP2 method. There is also evidence of wind scouring of snow at the site,
which may lessen the amplitude and resolution of historical variations in rBC deposition recorded
in the core. Air back-trajectory analyses suggest that, compared to Greenland, rBC deposition on
Devon ice cap is less sensitive to BC emissions from the North Atlantic sector (eastern North
America and western Europe) than Greenland is. We hypothesize that BC aerosols reaching Devon
ice cap originate more frequently from north-central/northwestern North America, and/or from
Russia and central Asia. The relatively long transport trajectories over the Arctic Ocean allow for
greater atmospheric mixing and deposition of aerosols to occur during transit, thus obscuring
source-receptor relationships. If correct, this interpretation implies that historical trends in BC
deposition over the Arctic, and the resulting albedo-climate forcing, are likely subject to large
spatial variability, even over the relatively short distance between Devon Island and Greenland.
This variability, which is probably linked to differences in BC aerosol transport patterns and
atmospheric residence time (Bauer et al. 2013), must be accounted for when attempting to model
the impact of past and future BC emission trends on the Arctic climate system.

This study also underscores the challenges of interpreting records of aerosol deposition

developed from firn or ice cores drilled on small ice caps or glaciers, where local topographic and
climatological effects can impact on the preservation of atmospheric signals, when compared with
the central regions of large ice sheets. A limitation of our study stems from the fact that the DV99.1
record of rBC deposition is from a different site than records of other aerosol species ($SO_4^{2-}$, Pb)
previously obtained from Devon ice cap summit. To verify our interpretation of the DV99.1 rBC
record, a new core should be drilled from the ice cap summit, or from another ice cap less affected
by wind scouring and melt-freeze effects (e.g., on northern Ellesmere Island), and on which co-
registered measurements of rBC and other aerosols could be made. This is particularly important
when one considers the large amount of spatial variability inherent in ice core records, even in
areas of optimal preservation (e.g., Gfeller et al., 2014).
**Acknowledgements**
The recovery of the DV99.1 ice core was supported by the Geological Survey of Canada and the
Polar Continental Shelf Project (Natural Resources Canada). Analysis of the core was funded by
Curtin University (Curtin Research Fellowship to R. Edwards, # RES-SE-DAP-AW-47679-1), and
by an Australian Endeavour Research Fellowship (# ERF_PDR_3051_2012) awarded to B.C.
Proemse. N. Schaffer (Univ. of Ottawa) assisted with some of the illustrations.

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

| Core | Lat. (N) | Lon. (W) | Max. depth (m) | Approx. elevation (m a.s.l.) | MAAT (°C) | Annual accum. (m H$_2$O) | Parameters measured |
|---|---|---|---|---|---|---|---|
| DV98.3 | 75.34° | 82.14° | 302 | 1930 | -12 | 0.25-0.28 | δ$^{18}$O, radioactivity, major ions, trace metals |
| DV99.1 | 75.32° | 81.64° | 169 | 1903 | | 0.16 | δ$^{18}$O, melt features, EC, rBC |
| DV2000 | 75.34° | 82.14° | 64 | 1930 | -12 | 0.25-0.28 | δ$^{18}$O, radioactivity, trace metals |

**Table 1.** Details of the three Devon ice cap cores used in this study. MAAT = Mean annual surface air temperature. See text for specific references to published data.

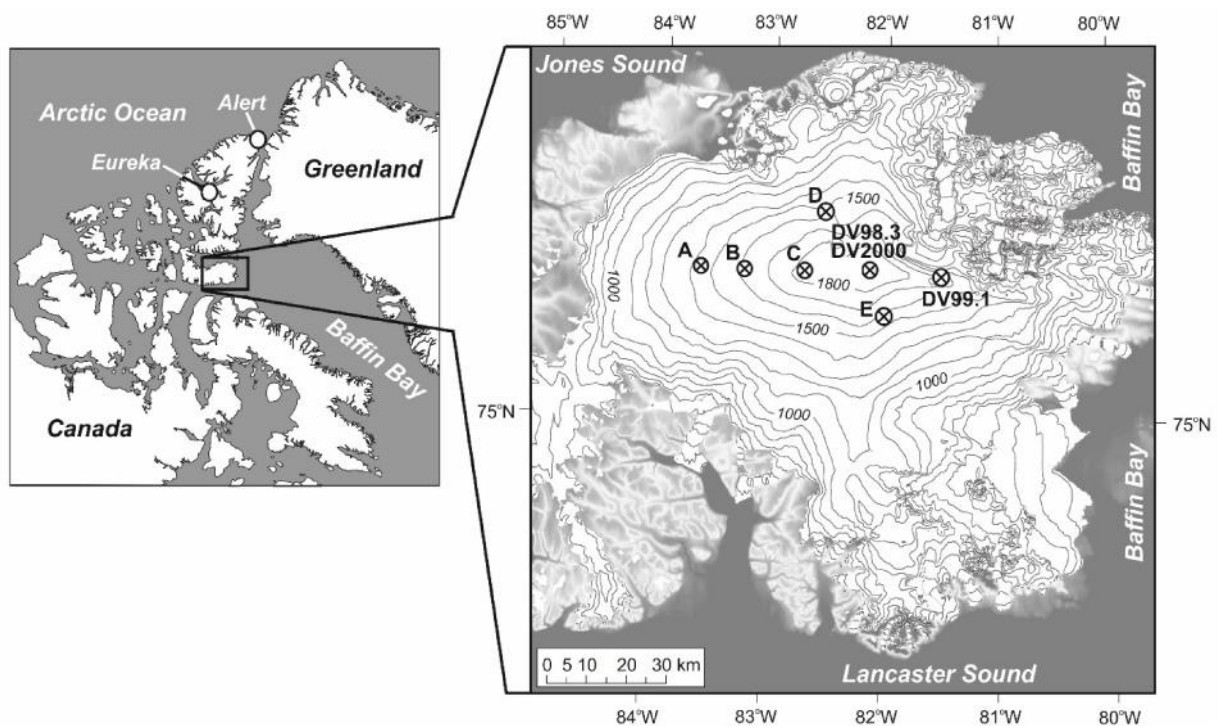

**Fig. 1.** Location map of the Canadian Arctic Archipelago (left), with enlargement of Devon ice (right). The location of the various ice core sites mentioned in the text are shown. Sites A to E refer to the shallow core array of Colgan and Sharp (2008). Elevation contours on Devon ice cap are spaced at 100 m above sea level.

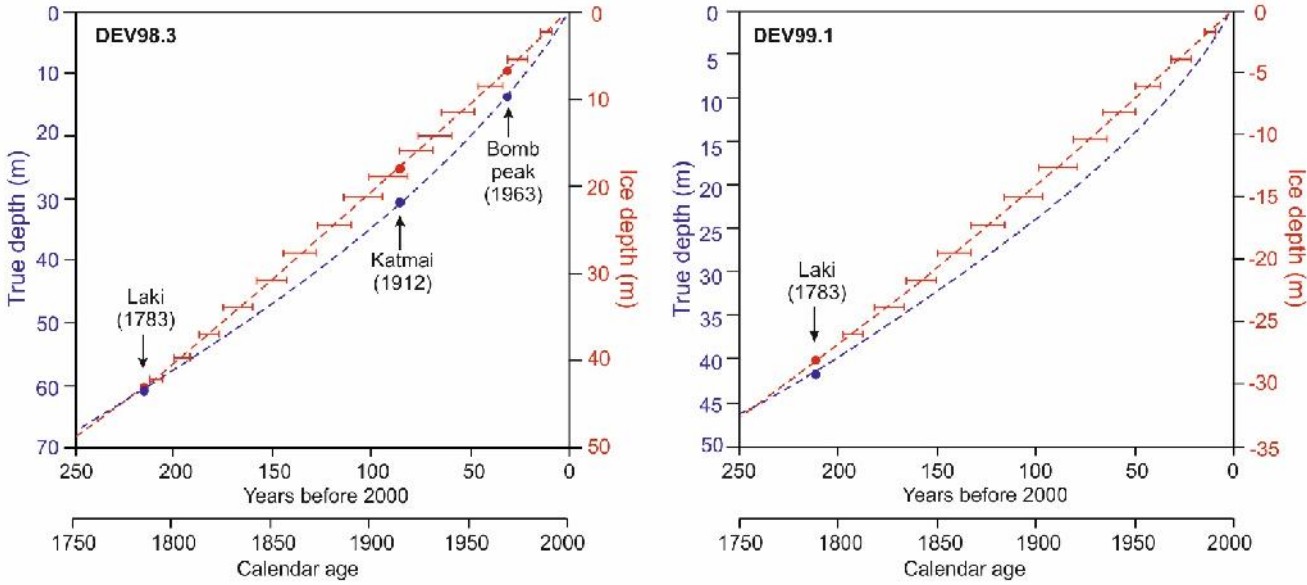

**Fig. 2.** Age models for parts of the DV98.3 and DV99.1 cores from Devon ice cap. The error bars on the age curve relative to ice-equivalent depths (red) bracket the 95 % confidence interval on the estimated age for discrete depths, as established from Monte Carlo simulations (see text).

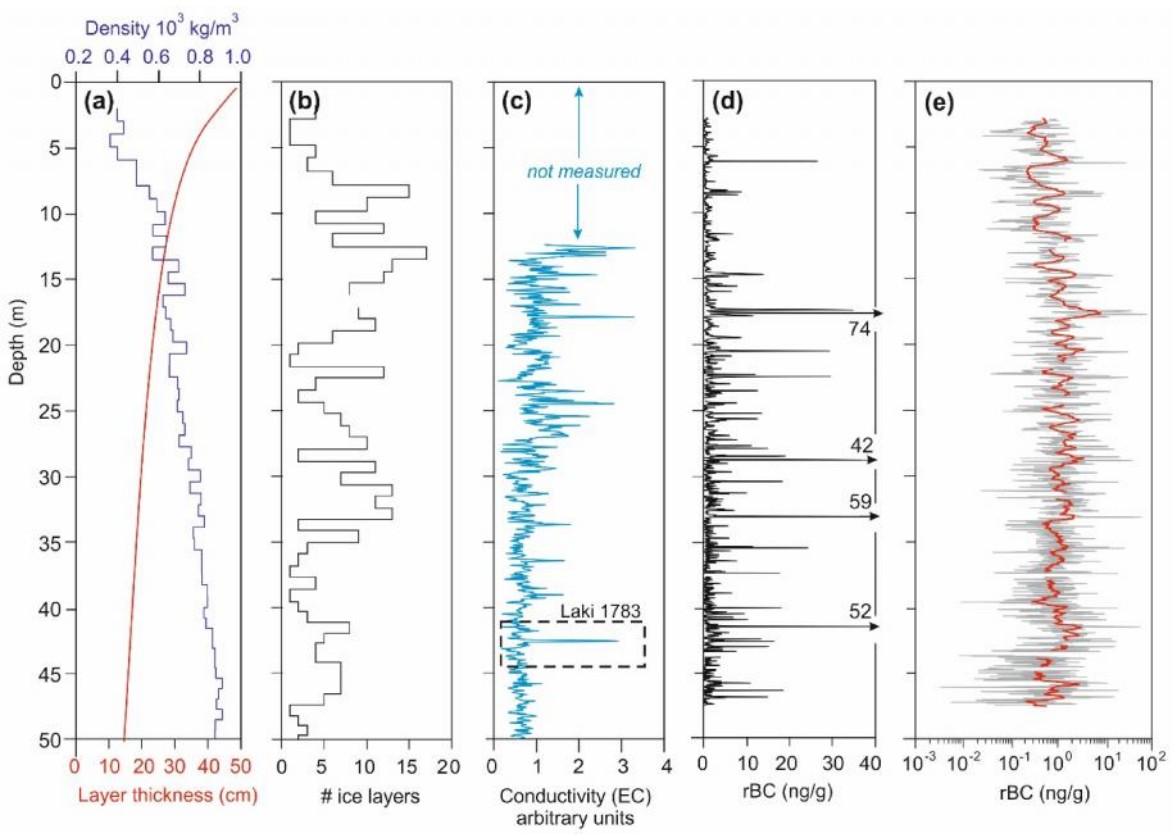

**Fig. 3.** Profiles of physical properties and rBC in the top 48-50 m of the DV99.1 ice core. **(a)** Firn density and estimated mean annual layer thickness. **(b)** Frequency of discrete ice layers (>3 mm thick) per core section. **(c)** Solid-state electrical conductivity (EC) of the core from 12.8 to 50 m depth. The EC peak attributed to acidic fallout from the Laki 1783 eruption is labelled. **(d)** and **(e)** rBC concentrations plotted on linear and log scales. The bold red line is a 500-point (~1-m) moving average.

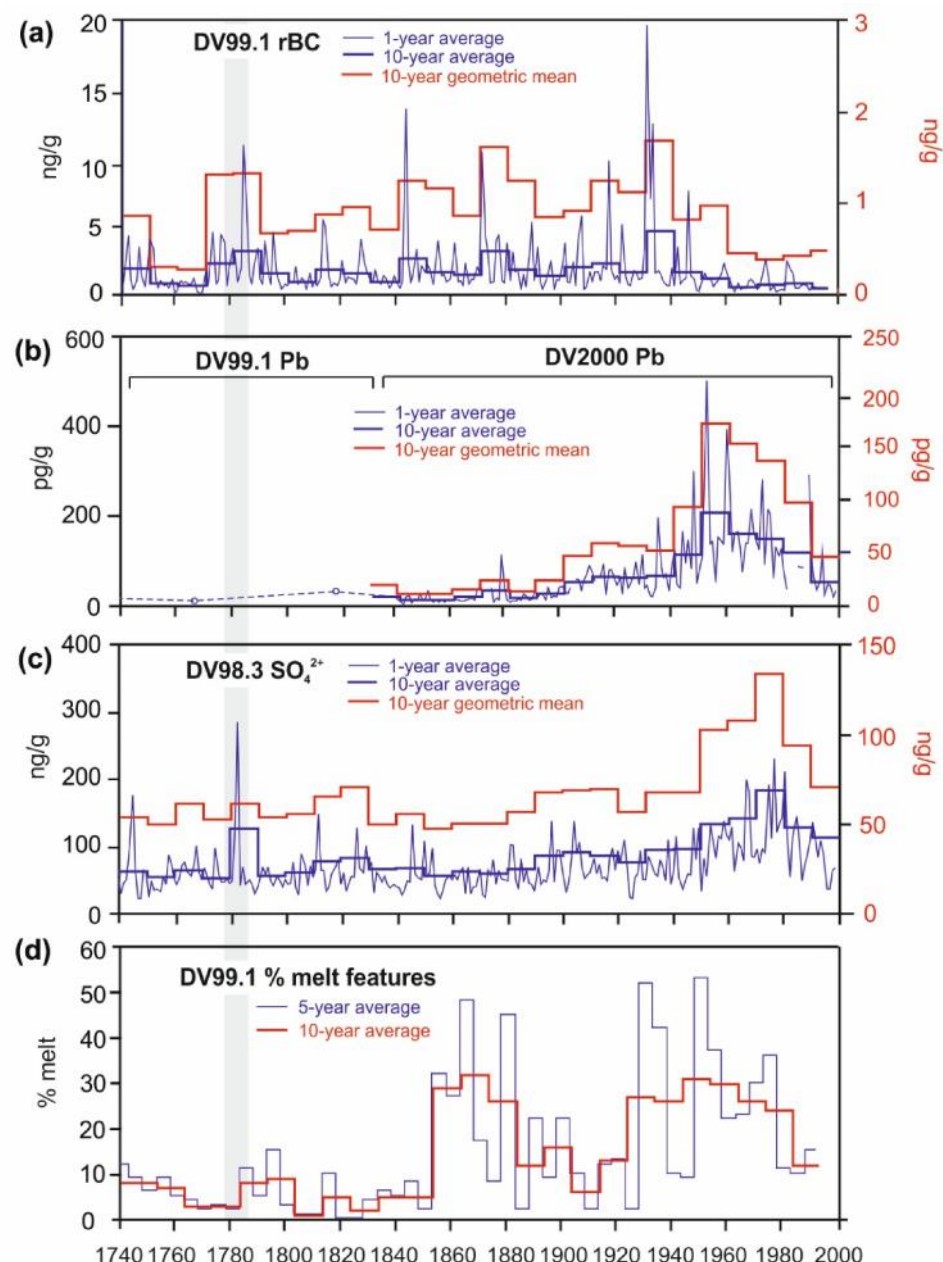

**Fig. 4.** Environmental changes on Devon ice cap, 1740-1999, recorded in three cores from the summit region (DV98.3, DV99.1 and DV2000). **(a)** rBC concentrations in the DV99.1 core; **(b)** Pb concentrations in the DV99.1 core (~1740-1840) and DV2000 core (1840-2000); **(c)** $SO_4^{2-}$ in the DV98.3 core; and **(d)** volumetric percentage of icy melt features in the DV99.1 core due to surface summer melt. Data are presented in ~1-, 5- and/or 10-year averages. For panels **(a)** to **(c)**, 10-year geometric mean values of the data are also plotted in red on separate scales (left). The shaded grey bar identifies the Laki 1783 isochron used to correlate the different cores. The width of the bar denotes the maximum dating uncertainty at the corresponding depths in these cores. The Pb data are from Shotyk et al. (2005) and Zheng et al. (2007), the $SO_4^{2-}$ data from Kinnard et al. (2006), and the melt feature data from Fisher et al. (2012).

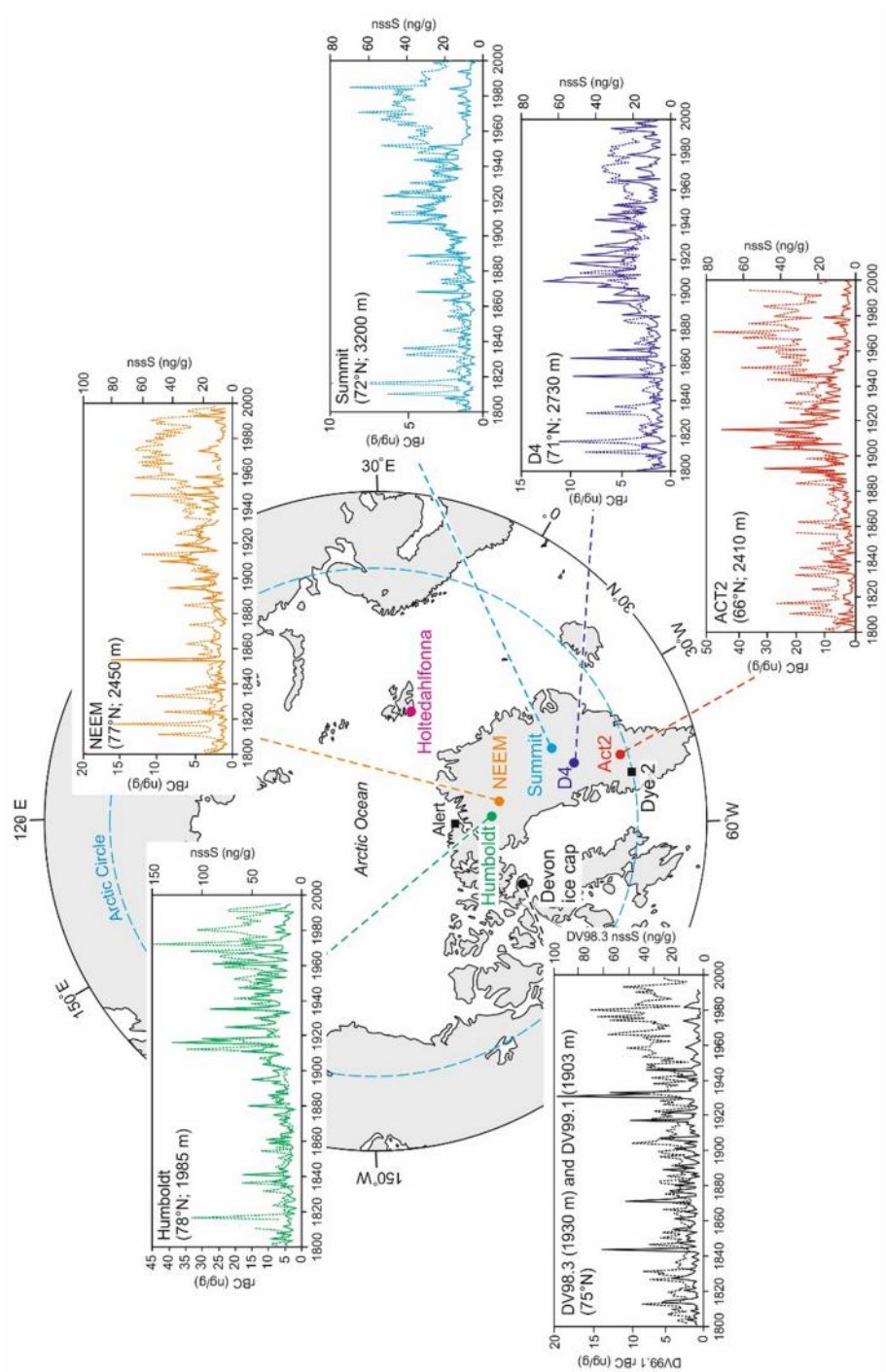

**Fig. 5.** The record of atmospheric rBC and non-sea salt sulfur (nssS) deposition on Devon ice cap over the period 1800-2000 compared with similar records developed at various sites in Greenland by identical or nearly-identical methods. Full lines are rBC; stippled lines are nssS. Data from Summit, D4, ACT2 and Humboldt: McConnell et al. (2007) and Koch et al. (2011); data from NEEM: Zennaro et al. (2014) and Sigl et al. (2015). Also shown is the location of the ice-core record of elemental carbon (EC) deposition developed from Holtedahlfonna, Svalbard, by Ruppel et al. (2014), as well as other sites (Alert, Dye 2) mentioned in the text.

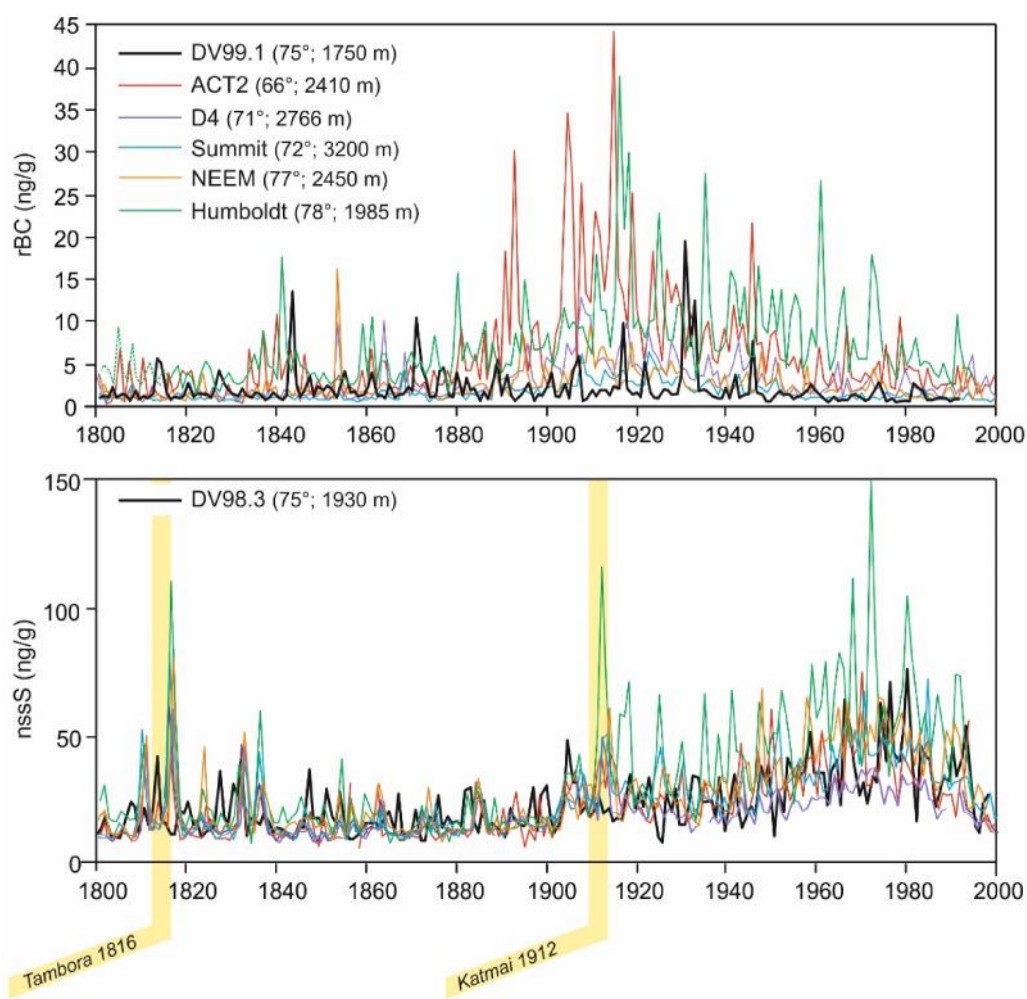

**Fig. 6. (a)** The DV99.1 record of atmospheric rBC deposition since 1800 compared with other records developed from sites in Greenland identified in **Fig. 5**. All records are presented in one-year averages. **(b)** As in (a) but for records of non-sea salt sulfur (nssS). Two volcanic eruption isochron used for correlation in the Greenland cores are highlighted.

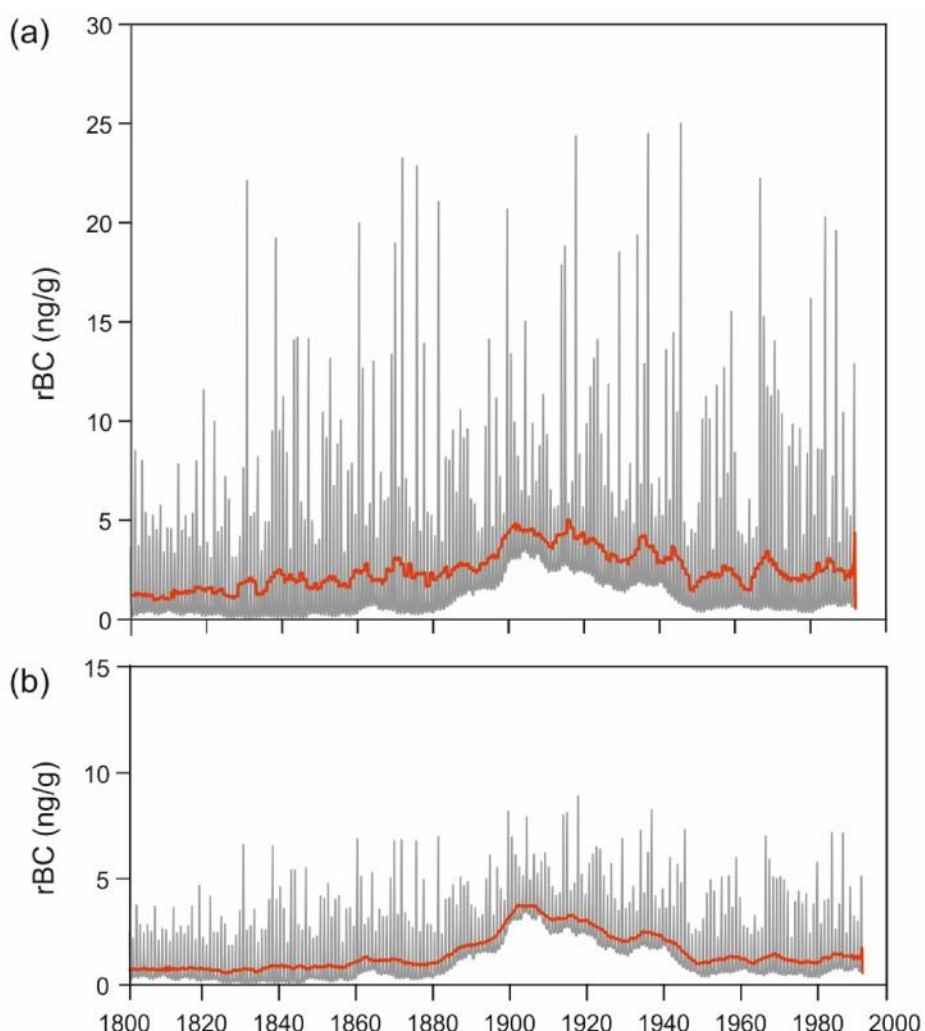

**Fig. 7.** Simulation of the effects of snow wind scouring on the preservation of an anthropogenic signal of rBC deposition in a synthetic ice-core times series of rBC spanning the period 1800-1990. **(a)** The synthetic series, with a pseudo-seasonal cycle superimposed on the interdecadal baseline trend observed in the Greenland D4 record (McConnell et al., 2007). **(b)** The synthetic series after randomly truncating the amplitude of all winter deposition peaks (November-March) by 30-60 %. The bold red line in both panels is a 5-year running geometric mean.

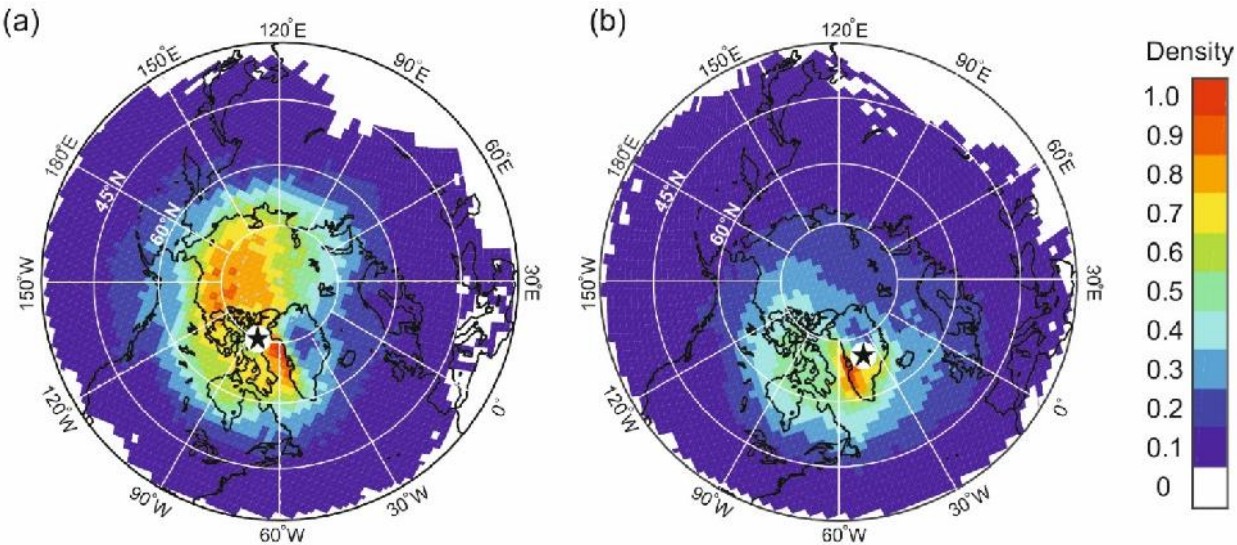

**Fig. 8.** Maps of residence time probability for air arriving at **(a)** Devon ice cap and **(b)** Summit, Greenland over the period 1948-1999, computed using HYSPLIT4. Air residence probability densities were normalized to a scale of 0-1, and were spatially detrended by multiplying the original residence time grids (in hours) by the distance between each grid point and the coring site. This effectively removes the concentric increase in probability density near the back-trajectory start point (Ashbaugh et al., 1985). The spatial resolution of the grid is 200 × 200 km.

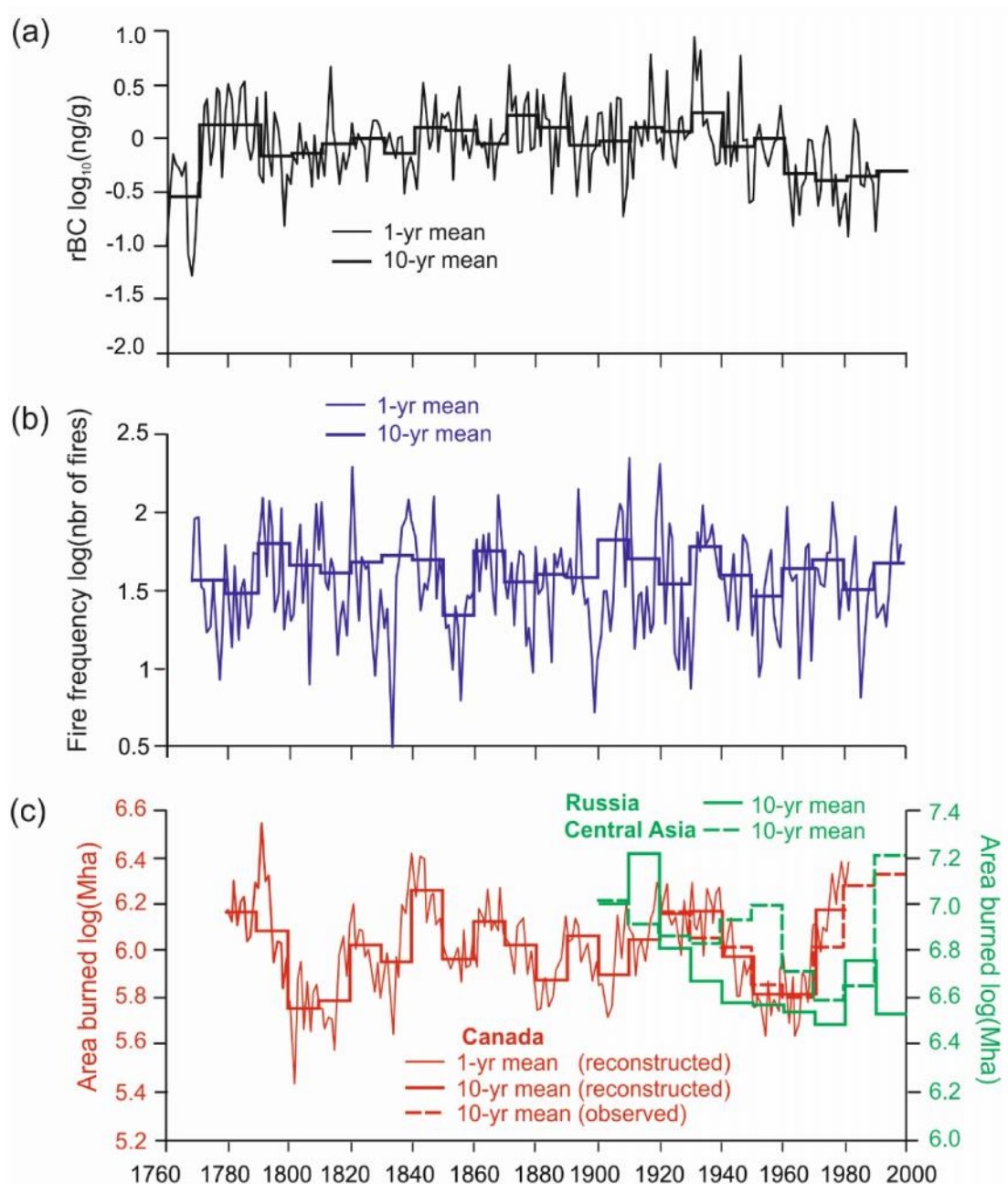

**Fig. 9. (a)** Historical variations in rBC concentration in the DV99.1 core, 1760-1992, compared with reconstructed historical trends in **(b)** fire frequency in the eastern boreal forest region of Canada (Girardin *et al.*, 2006), and **(c)** burned area across northern Canada (Girardin, 2007) and in the boreal and grassland regions of Russia and Central Asia (Mouillot and Field, 2005). All data were log-transformed to facilitate visual comparisons.