# Peer review of "Historical black carbon deposition in the Canadian High Arctic"

_Atmospheric Chemistry and Physics, 2017_

## Referee Comment (RC1) · Anonymous Referee #1 · 30 Oct 2017

The authors present valuable new BC data from the severely under-sampled Arctic. Historical BC records from different locations within the Arctic are essential for reliable model validation, and generally to understand the spatial and temporal variations in BC trends and processes affecting BC concentration and deposition patterns. Furthermore, discussion on post-depositional factors affecting BC records, such as wind scouring, is particularly welcome, as these are currently quite superficially known. The manuscript is well written and easy to follow. The used methods are appropriate, and the presented hypotheses are well justified and thoroughly discussed. I recommend publication if the following points are addressed in the revised manuscript.

[Figure]

Specific comments:

Generally, the authors are advised to more carefully choose citations.

Page 2, Line 14: Is the citation to Koch et al., 2011 and Lee et al. 2013 appropriate here, as these modelling papers don't present new ice core BC observations, as currently suggested by the sentence? For clarity, I'd suggest to replace these references with other appropriate work (e.g. McConnell 2010; Zennaro et al. 2014; Sigl et al. 2015) or modify the sentence. For instance, the Koch et al. (2011) cite McConnell et al. (2007), McConnell (2010) and McConnell and Edwards (2008) for the ice core observation data used in their modelling study.

P 6, l 27: Please, remove the citation to Koch et al. 2011, and replace e.g. with McConnell 2010 (if you like). Also, on P 6, l 28, Skeie et al. 2011 doesn't present any own BC emission inventory data, so please remove this citation as well.

P 12, l 12: The citation to Ruppel et al. (2014) is incorrect in this context. It should be replaced e.g. by Garrett et al. (2011) who study the issue of BC scavenging efficiencies during atmospheric transport to the Arctic (while Ruppel et al. (2014) only present hypotheses on the matter).

P 3, l 19-23: Could you clarify what these microparticles are, i.e. for instance give examples on what type of particles are meant?

Section 3.3 discusses impressively the uncertainties caused by stochastic spatial variations of deposition of the aerosols in snow, and post-depositional modifications (e.g., by wind scouring) in the ice core record, while actual measurement uncertainties of the nebulizer and SP2 instrumentation are not included. However, there is reason to believe that the chosen analysis methodology itself may cause significant uncertainties for the rBC record as well. As the authors discuss in the last paragraph of Section 4.2 (P8, l 18-31) it is known that the used nebulizer doesn't effectively aerosolize larger rBC particles which however may constitute a notable part of the total BC in the studied ice core, as it is shown to be affected by post-depositional processes (e.g. summer melt) which are known to increase the BC particle size in snow (e.g. Schwarz et al., 2013). The manuscript should therefore determine clearly which size fraction of rBC is quantified here. It is understandable why the nebulization and SP2 quantification efficiency is only discussed in Section 4.2, after discussing summer melt of the ice core. However, these uncertainties should at least be mentioned (if they are too difficult to be quantified) already in Section 3.3, and Section 4.2 referred for further discussion. Currently, in Section 3.3 the reader is erroneously led to believe that all uncertainties of the BC record are unrelated to the used rBC quantification methodology.

Finally, it would be good if the authors would present a total estimate (in numbers, %) of how large the uncertainties of the results are. P 5, l 27-28 says "Because the magnitude of $\varepsilon$s is independent from that of errors that arise from ice core dating uncertainties ($\varepsilon$t), the combined uncertainty was calculated as the quadratic sum of these terms ($\pm 2\sigma$).". Is the uncertainty seen in any of the figures? How large is the combined uncertainty? If these uncertainties ($\varepsilon$s and $\varepsilon$t) cannot be combined with the analytical method uncertainties, it would be good if these uncertainty percentages were presented separately (currently no uncertainty is given for the analytical methodology).

P 12, l 16: For clarity, I'd suggest the following addition (in parenthesis): "The time series differs from the Greenland records (measured with the same analytical methodology as used here) in that it. . .". The addition could clarify that the differences of the Devon Island BC record to the Greenland records are surprising as they are analyzed with the same methodology, while differences to the Svalbard record may be expected due to different methodology.

Technical corrections:

P 3, l 13: Consider replacing "sticks" with subsamples.

P 9, l 26: Replace "Their" with "their".

P 10, l 2: Remove comma ( , ) before the reference.

---

## Referee Comment (RC2) · Anonymous Referee #2 · 6 Nov 2017

Review Zdanowicz et al.: Historical black carbon deposition in the Canadian High Arctic: A 190-year long ice-core record from Devon Island

General Comments:

The authors present a reconstruction of black carbon and microparticle concentrations from an ice core from Devon Ice Cap in Canada covering the time period 1810-1990 AD. Where direct observations of atmospheric BC are scarce and limited to the most recent decades, ice-cores can – in principle – act as surrogates for direct observations and provide valuable information on the composition of the pre-industrial atmosphere and serve as benchmarks to assess the capabilities of models to realistically simu-

late the aerosol life-cycle and resulting forcing of past climate. In order to use such proxy reconstructions such glacio-chemical records need to realistically represent the atmospheric impurity content through time. In this respect, the current manuscript falls short in providing sufficient evidence for the reasons summarized below: The authors provide virtually no information that would allow to assessing their ability to achieve reproducible BC concentrations from ice cores or to repeat their experiments. They fail to report how they calibrated their measurements and omit to discuss any metrics (e.g., detection limit, stability, linearity, stability, repeatability, reproducibility) commonly considered necessary when introducing new instrumentation in analytical chemistry (see for example (Lim et al., 2014;Wendl et al., 2014;Mori et al., 2016;Bigler et al., 2011).

The same is true for the age-scales. The entire dating depends critically on the correct identification of the Laki signal in 1783 AD at this specific ice-core site and also for the other ice cores from Devon, but I can find in none of their cited papers a graph showing the full EC or SO4 record used to make this attribution. The same is true for the proxy signature of the 1963 AD nuclear bomb testing fallout. With no electrical or glacio-chemical signature of Laki provided in the manuscript and with having a huge sulfate spike recorded in 1847 AD which is not recorded in any other ice core from nearby Greenland, I consider it equally likely that the latter signal may as well be from Laki 1783, and your timescale off by over 60 years.

I consider it very unfortunate and not sufficiently well explained why you chose to limit your analyses to 1810-1990 AD and to only two new parameters. As stated above, Laki is crucial for the depth-age scale; the past 10 years would allow to have overlap with aerosol observations (e.g. from Alert); and analyzing additional aerosols in DV99.1 would allow you to 1) assess the effects of melting on your impurity records, 2) improve the relative dating among the different ice cores and timescales, and 3) to attribute specific sources to BC using for example NH4+ and SO42- as unique source tracers. The fact that the ice is fractured (>38m) or not consolidated (>4m) does not make measurements impossible, and half of an ice-core minus 2.5 x 2.5 cm consumed for your

[Figure]

CFA measurements should provide you with enough material for additional analyses.

Specific Comments:

Page 2: L. 25: How deep was the core, did you reach bedrock. Is this the same 170.6m long core as described by (Zheng et al., 2007) as D1999 core?

Page 3: L. 6: Surprising to see half of the core consumed for EC measurements and low resolution d18O analyses. What is the diameter of the cores? L. 9: What is the reference for the initial age estimate? L. 10-11: Zheng et al., (2007) report that the core quality was good for the entire core D1999? Is this the same core? If so, which statement is correct? Why don't you add a table with all meta data for ice cores and analyses you discuss in your manuscript? L.10: Why should unconsolidated snow not be useable for analyses? It is virtually impossible to contaminate with BC and analyses could have easily been performed with an SP2 on discrete samples. L12: When did the analysis take place? Over how many days, weeks, months? Did you observe sublimation on the ice surface after >15 years of storage? L13: What was class100? The cold room? The lab space? L12-14: I am missing references and I have never heard of an Advanced Ultra-clean Environmental facility. Is this the first time you are performing this kind of analyses in this lab? L14: At which melt rates did you melt the ice? How do you assure the flowrate is constant? It must be difficult with the frequent change of sold ice lenses (40-60% on average) and soft firn. If the flowrate is not constant, how do you correct for this? L14-21: Provide a chart with the analytical setup of all instruments. Provide information on calibration (standard material, linearity, stability) and reproducibility of the results. L.17-18: These citations are all for a lab in the USA L.20: How does the microparticle content connect to your scientific problem? What is the motivation? L. 24-25: Why did you not analyze these other aerosol species directly in DV99.1? There should be half of a core minus 2.5 x 2.5 cm of cross section left. This would allow you attribute with more confidence sources to biomass burning and coal burning. Comparisons to other ice cores drilled at different sites only allow you to compare some general trends. According to your age model you have but one

common age marker between these ice-core records (the alleged 1783 signal), strong spatial gradients in accumulation caused by wind erosion and/or melting.

Page 4: L. 4-12: Is any of this age models published? If so, please add the citation and the timescale name for the ice cores, respectively. It appears 2 of 3 citations at the end of this section are based on Agassiz ice cap. I do not find any figure showing the signatures attributed to Laki and Katmai in Kinnard et al. (2006). L. 4-12: Since your study is critically dependent on the correct identification of the two reference horizons 1783 and 1963, I expect to see all the data that was used for these attributions. It is very plausible that very large acidity in the Arctic were caused by the Laki eruption (Kekonen et al., 2005) but there may also other large acid layers recorded in the Arctic in e.g., 1765, 1815 (Wendl et al., 2015). Equally, the Arctic nuclear fallout signals are in general much broader (1954-1963, (Arienzo et al., 2016)) than described here. How sharp is your signal compared to these other ice cores? Maybe this could tell you something about potential redistribution of impurities caused by melting. L. 15: Provide these EC measurements for the DV99.1 ice core. How reliable is this peak, if it was recorded in the fractured ice-core sections >38m. Was it reproduced by sulfate analyses? L. 17: What accumulation rates to you get between year of drilling, 1963 and 1783 for each of the DV ice cores? Consider providing this information in supplement. L. 24-26: Please show these common signatures from DV99.1, DV98.1, DV98.3 L. 26: What do you mean with adopted? How much depth-age models exist? Where are they published? Are there any isochrones between these age-models and ice cores? L. 28: How much meters apart were DV98.3 and DV2000 drilled? Is it appropriate to use the same chronology for two different cores given that snow fall and snow conservation on summits are varying on very small spatial scales. I would only adopt a timescale for another ice core if this was supported by a number of isochrones. L. 31: I doubt the true effective resolution of the measurements is at a mm scale. You may be recording data at a rate equivalent to mm in depth, but you need to account for the uncertainty in the depth registration and dispersion of the signals through mixing (Bigler et al., 2011).

[Figure]

Page 5: L. 4-12: Add analytical uncertainties as well. L. 6: (e.g. wind scouring, melt induced relocation). According to Kinnard et al. (2006) average melt rates in D99 are 50% after 1850 AD, which appears to me a significant factor that could modify the impurity records one way or the other. L. 9-18: This approach assumes the layer thickness variation is the only source of uncertainty in estimating the age error. It assumes the Laki event is correctly attributed, and must take into account some prior knowledge of the snow accumulation rates and its variability. I don't see how you can estimate the variability "between reference layers of known age" without being able to count annual layers.

Page 6: L. 17-18: Microparticle concentrations do not follow the same trend than nssS and Pb but have a clear step-function. On which observation do you base your attribution of "anthropogenic pollution"? L. 22: Such source attributions would strongly benefit of having all parameters analyzed on the same core. L. 23: Black carbon concentrations

Page 7: L. 12: Necessary not only legitimate L. 14-30: I agree on this point.

Page 8: L. 6-17: Are the sections in the ice with the increased melt layer occurrence believed to be the periods experiencing more melting? Or are they accumulating the meltwater (plus impurities) from the ice sections above? In other words: how deep does percolation go? Is there surface runoff carrying impurities away? Given that DV is only 700 km away from NEEM and Humboldt ice cores and all ice cores agreeing on showing strong BC deposition in early 20th century I tend to believe the differences in BC deposition is not from differences in atmospheric burden, but from some aspect specific to the Devon ice cap. The low elevation and observed melt features appear to make a strong case that the impurities along the ice core may be subject to severe loss and/or redistribution in particular during warmer time periods (e.g. Arctic warming 1920-40s, (Yamanouchi, 2011)), which would smear and bias any atmospheric information. L. 18-28: Post-depositional coagulation moving BC sizes out of the detectable range also seems a very plausible explanation for reduced BC recovery during especially warm periods. Low reproducibility of replicate measurements when samples were subject to melting and freezing cycles is reported by several research groups performing BC analyses in ice and snow (apparent "loss" rates in the order of 50%); as a result performing BC analyses on samples that have been refrozen is strongly discouraged (see e.g. (Lim et al., 2014;Wendl et al., 2014)).

Page 9: L. 2-9: This may be true if you compared Devon Ice Cap with Central and Southern Greenland ice cores, but NEEM and Humboldt are just 700-800km away from Devon and are thought to have largely similar source regions for aerosols and precipitation (Zennaro et al., 2014).

Page 10: L. 2-26: How meaningful is such a comparison given the low degrees of free-dom (resulting from decadal data), dating uncertainties and the inability to differentiate industrial from BB BC in the Devon ice core? Are these correlation stables if you varied binning, removed the common declining trend over most of the 20th century? L. 29-31: This is not surprising; as you outline below: K+, NH4+ and BC have multiple sources and probably also different chemical properties making them more or less susceptible to melt-induced relocation.

Page 11: L. 2: Could these low numbers for the most recent time period indicate some loss from melting caused by the rapid warming of the Arctic? To my knowledge and supported by Figs. 5 and 6 concentrations of rBC lower than during the pre-industrial baseline are not recorded for any other ice-core in the Arctic. L. 15: Given the potential limitations from inadequate nebulization, potential loss and redistribution of impurities, I strongly doubt that this value is a realistic approximation of the true atmospheric BC influx. L. 25: Note that you use the same abbreviation EC for both electrical conductivity and elemental carbon.

Page 12: L1-2: Does Humboldt show similar melt features than Devon Ice cap? The agreement between Humboldt and the other Greenland records (in both BC and nssS) seems very high. L9-12: None of the emission inventories or any ice core suggests

that mean BC emissions from 1960-1990 were below preindustrial (i.e. before 1850 AD) levels, as the Devon ice-core seems to imply.

Figures: Fig. 4: Extend the x-axis to include your only reference marker in 1783 AD. Units in panel b are missing and it is nssSO42-. I do not see any signature related to the largest VEI=6 eruptions of Katmai, Krakatao, Tambora and 1809, but a huge SO4 signal around 1847 AD. Do you have any explanation? How do you know this is not from Laki 1783 or Tambora 1815? How do you calculate K+BB? Fig.4 and Fig.6: DV98.3 nssSO4 (panel b, Fig 4) appears to be different from nssS (Fig. 6), the latter is peaking in the 1960-1980s, the first starts to peak only after 1980. Which one is correct? Fig. S3: Check the lower panel. There should be only one y-value for a given depth value

Technical Corrections: Page 2: L. 4: atmospheric chemistry climate models L. 14: add e.g. since the list is incomplete L22: ice core were previously drilled Page 3: L 29-30: Na+ (twice)

References

Arienzo, M. M., McConnell, J. R., Chellman, N., Criscitiello, A. S., Curran, M., Fritzsche, D., Kipfstuhl, S., Mulvaney, R., Nolan, M., Opel, T., Sigl, M., and Steffensen, J. P.: A Method for Continuous (PU)-P-239 Determinations in Arctic and Antarctic Ice Cores, Environ Sci Technol, 50, 7066-7073, 10.1021/acs.est.6b01108, 2016.

Bigler, M., Svensson, A., Kettner, E., Vallelonga, P., Nielsen, M. E., and Steffensen, J. P.: Optimization of High-Resolution Continuous Flow Analysis for Transient Climate Signals in Ice Cores, Environ Sci Technol, 45, 4483-4489, Doi 10.1021/Es200118j, 2011.

Kekonen, T., Moore, J., Peramaki, P., and Martma, T.: The Icelandic Laki volcanic tephra layer in the Lomonosovfonna ice core, Svalbard, Polar Res, 24, 33-40, DOI 10.1111/j.1751-8369.2005.tb00138.x, 2005.

[Figure]

Lim, S., Fain, X., Zanatta, M., Cozic, J., Jaffrezo, J. L., Ginot, P., and Laj, P.: Refractory black carbon mass concentrations in snow and ice: method evaluation and intercomparison with elemental carbon measurement, Atmos Meas Tech, 7, 3307-3324, 10.5194/amt-7-3307-2014, 2014.

Mori, T., Moteki, N., Ohata, S., Koike, M., Goto-Azuma, K., Miyazaki, Y., and Kondo, Y.: Improved technique for measuring the size distribution of black carbon particles in liquid water, Aerosol Sci Tech, 50, 242-254, 10.1080/02786826.2016.1147644, 2016.

Wendl, I. A., Menking, J. A., Farber, R., Gysel, M., Kaspari, S. D., Laborde, M. J. G., and Schwikowski, M.: Optimized method for black carbon analysis in ice and snow using the Single Particle Soot Photometer, Atmos Meas Tech, 7, 2667-2681, 10.5194/amt-7-2667-2014, 2014.

Wendl, I. A., Eichler, A., Isaksson, E., Martma, T., and Schwikowski, M.: 800-year ice-core record of nitrogen deposition in Svalbard linked to ocean productivity and biogenic emissions, Atmos Chem Phys, 15, 7287-7300, 10.5194/acp-15-7287-2015, 2015. Yamanouchi, T.: Early 20th century warming in the Arctic: A review, Polar Sci, 5, 53-71, 10.1016/j.polar.2010.10.002, 2011.

Zennaro, P., Kehrwald, N., McConnell, J. R., Schupbach, S., Maselli, O. J., Marlon, J., Vallelonga, P., Leuenberger, D., Zangrando, R., Spolaor, A., Borrotti, M., Barbaro, E., Gambaro, A., and Barbante, C.: Fire in ice: two millennia of boreal forest fire history from the Greenland NEEM ice core, Clim Past, 10, 1905-1924, 10.5194/cp-10-1905-2014, 2014.

Zheng, J. C., Shotyk, W., Krachler, M., and Fisher, D. A.: A 15,800-year record of atmospheric lead deposition on the Devon Island Ice Cap, Nunavut, Canada: Natural and anthropogenic enrichments, isotopic composition, and predominant sources, Global Biogeochem Cy, 21, Artn Gb2027 10.1029/2006gb002897, 2007.

[Figure]

2017.

---

## Author Comment (AC1) · 31 Jan 2018

**Authors' response to referee comments**

**Anonymous Referee #1**

**General Comments:**

The authors present valuable new BC data from the severely under-sampled Arctic. Historical BC records from different locations within the Arctic are essential for reliable model validation, and generally to understand the spatial and temporal variations in BC trends and processes affecting BC concentration and deposition patterns. Furthermore, discussion on post-depositional factors affecting BC records, such as wind scouring, is particularly welcome, as these are currently quite superficially known. The manuscript is well written and easy to follow. The used methods are appropriate, and the presented hypotheses are well justified and thoroughly discussed. I recommend publication if the following points are addressed in the revised manuscript.

**Specific comments, and authors' responses:**

Page 2, Line 14: Is the citation to Koch et al., 2011 and Lee et al. 2013 appropriate here, as these modelling papers don't present new ice core BC observations, as currently suggested by the sentence? For clarity, I'd suggest to replace these references with other appropriate work (e.g. McConnell 2010; Zennaro et al. 2014; Sigl et al. 2015) or modify the sentence. For instance, the Koch et al. (2011) cite McConnell et al. (2007), McConnell (2010) and McConnell and Edwards (2008) for the ice core observation data used in their modelling study.

*P* 6, I 27: Please, remove the citation to Koch et al. 2011, and replace e.g. with McConnell 2010 (if you like). Also, on *P* 6, I 28, Skeie et al. 2011 doesn't present any own BC emission inventory data, so please remove this citation as well.

Changes in the citations were made as suggested.

P 12, I 12: The citation to Ruppel et al. (2014) is incorrect in this context.

**Removed.**

*P* 3, I 19-23: Could you clarify what these microparticles are, i.e. for instance give examples on what type of particles are meant?

We decided to remove the microparticle data from the revised manuscript, and focus solely on the rBC data.

Section 3.3 discusses impressively the uncertainties caused by stochastic spatial variations of deposition of the aerosols in snow, and post-depositional modifications (e.g., by wind scouring) in the ice core record, while actual measurement uncertainties of the nebulizer and SP2 instrumentation are not included. However, there is reason to believe that the chosen analysis methodology itself may cause significant uncertainties for the rBC record as well. As the authors discuss in the last paragraph of Section 4.2 (P8, I 18-31) it is known that the used nebulizer doesn't effectively aerosolize larger rBC particles which however may constitute a notable part of the total BC in the studied ice core, as it is shown to be affected by post-depositional processes (e.g. summer melt) which are known to increase the BC particle size in snow (e.g. Schwarz et al., 2013). The manuscript should therefore determine clearly which size fraction of rBC is quantified here. It is understandable why the nebulization and SP2 quantification efficiency is only discussed in Section 4.2, after discussing summer melt of the ice core. However, these uncertainties should at

least be mentioned (if they are too difficult to be quantified) already in Section 3.3, and Section 4.2 referred for further discussion. Currently, in Section 3.3 the reader is erroneously led to believe that all uncertainties of the BC record are unrelated to the used rBC quantification methodology.

Uncertainties in the SP2 measurements, and from other sources, are now discussed in section 3.3, with additional material on the analytical data and quality control in the supplement to the revised article. The size range of rBC particles measured by the SP2 instrument in the DV99.1 core is unfortunately not know at present.

Finally, it would be good if the authors would present a total estimate (in numbers, %) of how large the uncertainties of the results are. P 5, 127-28 says "Because the magnitude of  $\varepsilon_s$  is independent from that of errors that arise from ice core dating uncertainties ( $\varepsilon_t$ ), the combined uncertainty was calculated as the quadratic sum of these terms ( $\pm 2\sigma$ ).".Is the uncertainty seen in any of the figures? How large is the combined uncertainty? If these uncertainties ("s and "t) cannot be combined with the analytical method uncertainties, it would be good if these uncertainty percentages were presented separately (currently no uncertainty is given for the analytical methodology).

See response to above comment. On Fig 4 of the revised manuscript, we chose to remove the confidence limits, because they overcrowded the plots and made trends in the data (where they occur) difficult to see. However, we added plots of estimates for different sources of uncertainties (due, for e.g., to age model errors) in the supplement to the revised article.

*P* 12, *I* 16: For clarity, *I'd* suggest the following addition (in parenthesis): "The time series differs from the Greenland records (measured with the same analytical methodology as used here) in that it. . .". The addition could clarify that the differences of the Devon Island BC record to the Greenland records are surprising as they are analyzed with the same methodology, while differences to the Svalbard record may be expected due to different methodology.

The revised text as been modified as suggested.

**Anonymous Referee #2**

**General Comments:**

The authors present a reconstruction of black carbon and microparticle concentrations from an ice core from Devon Ice Cap in Canada covering the time period 1810-1990 AD. Where direct observations of atmospheric BC are scarce and limited to the most recent decades, ice-cores can - in principle - act as surrogates for direct observations and provide valuable information on the composition of the pre-industrial atmosphere and serve as benchmarks to assess the capabilities of models to realistically simulate the aerosol life-cycle and resulting forcing of past climate. In order to use such proxy reconstructions such glaciochemical records need to realistically represent the atmospheric impurity content through time. In this respect, the current manuscript falls short in providing sufficient evidence for the reasons summarized below: The authors provide virtually no information that would allow to assessing their ability to achieve reproducible BC concentrations from ice cores or to repeat their experiments. They fail to report how they calibrated their measurements and omit to discuss any metrics (e.g., detection limit, stability, linearity, stability, repeatability, reproducibility) commonly considered necessary when introducing new instrumentation in analytical chemistry (see for example (Lim et al., 2014; Wendl et al., 2014; Mori et al., 2016; Bigler et al., 2011). The same is true for the age-scales. The entire dating depends critically on the correct identification of the Laki signal in 1783 AD at this specific ice-core site and also for the other ice cores from Devon, but I can find in none of their cited papers a graph showing the full EC or SO4 record used to make this attribution. The same is true for the proxy signature of the 1963 AD nuclear bomb testing fallout. With no electrical or glacio-chemical signature of Laki provided in the manuscript and with having a huge sulfate spike recorded in 1847 AD which is not recorded in any other ice core from nearby Greenland, I consider it equally likely that the latter signal may as well be from Laki 1783, and your timescale off by over 60 years.

I consider it very unfortunate and not sufficiently well explained why you chose to limit your analyses to 1810-1990 AD and to only two new parameters. As stated above, Laki is crucial for the depth-age scale; the past 10 years would allow to have overlap with aerosol observations (e.g. from Alert); and analyzing additional aerosols in DV99.1 would allow you to 1) assess the effects of melting on your impurity records, 2) improve the relative dating among the different ice cores and timescales, and 3) to attribute specific sources to BC using for example NH4+ and SO42- as unique source tracers. The fact that the ice is fractured (>38m) or not consolidated (>4m) does not make measurements impossible, and half of an ice-core minus 2.5 x 2.5 cm consumed for your CFA measurements should provide you with enough material for additional analyses.

**Authors' response:**

First, some context needs to be provided. This has no incidence on the revisions to the paper as such, but it may help explain to the reviewer some of the shortcomings identified in the study. The analysis of the DV99.1 core for rBC was, to some extent, part of a "rescue operation" of the Canadian ice-core collection which, in 2011, was in danger of being permanently moth-balled as a result of the shutting down of the GSC ice-core laboratory in Ottawa (what remains of the collection eventually found a new home at the University of Alberta in 2017). At the time, an open call was sent to researchers interested in obtaining cores for various analyses. One outcome was that parts of the DV99.1 core were shipped to Curtin University for rBC analysis. Under more favorable circumstances, the core would also have been sampled and analyzed for ionic chemistry by IC at the GSC ice-core laboratory, where a core melter-sampling system had been set up. This, however, was not possible, which explains why we do not have co-registered ionic chemistry data to accompany the rBC and microparticle data for this core. This is admittedly not optimal, and it

reflects the conditions under which the study was initiated. Since the submission and review, we have been able to use additional, previously unprocessed analytical results, and to extend the rBC record to 1735, spanning the period including the Laki 1783 volcanic marker. Extending the record required a large additional effort. These new data are now included in the manuscript, and the plots and discussion have been updated accordingly.

The description of methods in the revised manuscript has been expanded, and additional include additional details are provided about instrumental settings, calibration, etc. A discussion of uncertainities in the SP2 analyses has been added in section 3.3. Some specific questions regarding data quality are answered below.

A number of minor mistakes and discrepancies in the paper were also corrected in the revised version. For example, the elevation of the DV99.1 site was incorrectly reported as 1750 m a.s.l. This has been revised to 1903 m a.s.l., and the altitude of the true summit of Devon ice ca, where the DV98.3 core was recovered, is actually 1930 m a.s.l.. These figures, and the coordinates of the coring sites, were verified with a recent detailed digital elevation model of the ice cap, and the correct values are now listed in Table 1.

Reviewer # 2 identified shortcomings and/or inconsistencies in the discussion of the age models of the DV99.1 and DV98.3 cores. To address some of these issues, we contacted Dr David Fisher, formerly with the GSC (now retired), who initially developed the age models for these cores. Dr Fisher kindly helped to clarify matters. To acknowledge his contribution, we added him as a co-author of the revised version of the manuscript.

Briefly: whenever an age model for a core is being developed, several iterations of the model are tested by choosing various plausible candidate volcanic marker peaks in EC and/or  $SO_4^{2^\circ}$  profiles (as the case may be). The choice of the most plausible model is determined by several considerations. For e.g., the model should not result in changes in the age-depth relationship that would imply unrealistic changes in the mean accumulation rate at the coring site. And the model should be compatible with other information about conditions at the coring site such as the estimated snow accumulation rate obtained by radioactive measurements (when such data are available). Preference is also given to the model that provides the greatest coherence in the low-frequency variations of  $\delta^{18}O$  records between cores and coring sites, as these variations are expected to be coherent and synchronous at regional scales (even if details may vary). The inconsistencies in the age model descriptions given in the submitted paper arose from some confusion between two different iterations of the age model used for the DV98.3 core. This was clarified in our exchanges with D. Fisher.

A more thorough and explicit description of the age models, with additional references, is now included in the revised manuscript, and the accompanying supporting information. The key points are discussed below, in order to avoid unnecessary repetitions in our response to the reviewer's detailed comments.

The age model of the **DV99.1 core** was developed in the manner described in Zheng *et al.* (2007), using a combination of ice flow model, EC measurements to identify volcanic signatures, and, at greater depth, the characteristic  $\delta^{18}$ O step that marks the transition from glacial conditions into the early Holocene. The EC profile starts at a depth of 12.38 m: section of cores above this were of brittle firn that provided inadequate electrode contact for the hand-held instrument we used. An EC peak that was tentatively assigned to the Katmai 1912 eruption occurs at a depth of 25.50 m, but it is neither particularly large nor a truly distinctive peak, as it is superimposed on a "shoulder" in the baseline conductivity (starting at ~40 m depth) which likely corresponds to the rise in

acidifying aerosol deposition on Devon ice cap during the industrial era. It is known from other Canadian ice cores that this rise began ca 1870 on southern Baffin Island, and slightly later (ca 1900) on northern Ellesmere Island (Goto-Azuma et al., 2002; Goto-Azuma and Koerner, 2001), so it probably occurred sometime between these dates on Devon Island. We did not show the presumptive 1912 Katmai marker in the DV99.1 age model plot of the submitted version of the paper, as its identification is uncertain, but it was used nonetheless to constrain the age model. This has now been added to the revised age model plots. The Katmai 1912 signal stands much more clearly in ice cores from Penny ice cap on Baffin Island (Goto-Azuma et al., 2002). The EC peak in the DV99.1 core that was assigned to the Laki 1783 eruption is at a depth of 42.6 m (29.6 m ice-equivalent). It is the highest and sharpest peak in this part of the profile, and it clearly precedes the rise in baseline conductivity, just as it does in cores from Penny ice cap. There are no deeper EC peaks of comparable magnitude in this part of the DV99.1 core. There are a few shallower peaks between 33 and 40 m depth, but these are of lower magnitude, and could be ascribed to other eruptive events in the early 19th century, such as Tambora (1816), but they are not sufficiently distinctive to be confidently assigned to specific events. Hence we have no strong reasons to revise our assignment of the ECM peak at 42.6 m to Laki (1783). This identification is more reliable than that of the presumed Katmai 1912 EC marker at 25.5 m. Based on this assignment of the Laki 1783 marker, the mean accumulation rate at the DV99.1 site was estimated to be 0.14 m ice  $a^{-1}$  (~0.16 H2O  $a^{-1}$ ) over the period 1783-1992. (Note that in the submitted paper, the figure of 0.14 m a-1 was erroneously given as being in water equivalent).

The best fit of the ice-flow age model to the various volcanic (EC) and  $\delta^{18}$ O markers in the entire DV99.1 core gives a predicted depth of 41.6 m for the 1783 layer, which is offset my one meter from the actual depth (42.6 m) at which the EC peak assigned to Laki occurs. This offset falls within plausible dating error limits if one allows for random interannual variations in the local snow accumulation rate. There is a larger (3.1 m) discrepancy between the model-predicted depth for Katmai 1912 (21.3 m) and the EC peak assigned to it (24.5 m), but as the identification of this EC marker is tentative at best, this discrepancy may not be meaningful. The manner in which the uncertainty of the age models were estimated (for both the DV99.1 and DV98.3 cores) is described below under our response to the reviewer's comment about page 5, L9-18.

The age model of the 302-m long **DV98.3** core (drilled near the true summit of Devon ice cap) was previously described in Kinnard et al. (2006) and also, more succinctly, in Krachler et al. (2005). No EC measurements were performed on this core, but there were measurements of radioactivity as well as high-resolution measurements of  $\delta^{18}$ O and ionic chemistry data, including SO42-. The radioactivity profile in firn/ice was measured by total beta ( $\beta$ ) activity in the borehole, as well as by tritium (3H) concentration and 137Cs activity in the core itself. The 3H profile can be seen in Clark et al. (2007), and the  $\beta$  and 137Cs profiles in Pinglot et al. (2003). All of these profiles agree very well and show two clearly-separated peaks, the higher and shallower one centered at a depth of 16.6 m below the 1998 surface. This was assigned an age of 1963, in accordance with the chronology of Dibb et al. (1992) and Kudo et al. (1998) for radioactive fallout in the Arctic. The deeper, smaller peak is assumed to correspond to the earlier, mid-/late 1950s peak in radioactive fallout. The reviewer asked whether the radioactive peak in the DV98.3 core could be used to assess the effect of post-depositional melt on the glaciochemical records in this core. This analysis was in fact already done and reported by Pinglot et al. (2003), who derived a proxy melt index from the radioactive profiles in Arctic ice cores. This index should be equal to 1 in the absence of post-depositional changes in radionuclide distribution in snow and firn, and increase with higher melt rates. For the DV98.1 core site, the derived index value was 1.2. For comparison, it was estimated to be 2 at low-elevation Svalbard ice coring sites, where summer melt rates are far higher than on Devon ice cap. Comparison of the radioactivity profiles in the DV98.3 borehole/core and the 210Pb activity profile at Summit, Greenland (Dibb et al., 1992), shows very good agreement, suggesting limited post-depositional "smearing" at the DV98.3 site, although the

possibility remains that the position of the peaks at DV98.3 could have been slightly offset by melt relative to their initial stratigraphic location. (See also discussion on estimates of melt-percolation effects, below)

Based on the 1963 radioactive peak age assignment, the estimated mean accumulation rate at the DV98.3 coring site over the period 1963-1998 is ~0.25 ± 0.2 m ice a-1 (Kinnard *et al.*, 2006; Clark *et al.*, 2007). Seasonal or pseudo-seasonal variations in the  $\delta^{18}$ O and SO42- profiles are only recognizable in the uppermost part of the DV98.1 core (top ~15 m). Hence their usefulness for annual layer identification is limited. However, performing a cross-spectral analysis of these data (normalized to ice-equivalent depths) allows us to identify the frequencies at which most of the spectral power is concentrated. Applying this approach to the top ~11 m of the DV98.3 core (5 m in ice equivalent) reveals an unambiguous spectral peak at a frequency of 0.042 cm-1, which corresponds to an approximate annual layer thickness of 0.24 m ice-equivalent, or an annual mean net accumulation rate of 0.26 m H2O a-1 for the most recent decades in the record (see Kinnard *et al.*, 2006). In addition, there are over 50 years of continuous annual winter mass balance measurements available from the summit of Devon ice cap, and these constrain the mean accumulation rate there to 0.25-0.28 m H2O a-1 for the latter part of the 20th century.

As described in Kinnard et al. (2006) one of the SO42 peak that was used to constrain the age model for the DV98.3 occurs at an ice-equivalent depth of 25.5 m, and this was assigned to the 1912 eruption of Katmai. Using this marker gives an estimated mean accumulation rate for the 1912-1998 period of 0.29 m ice  $a^{-1}$ , or 0.32 m H2O  $a^{-1}$ . Two other large SO42- peaks occur at 61.55 and 81.55 m depths, respectively. The shallower one is narrow and sharp, the deeper of lesser amplitude and much broader. One of these peaks likely represents fallout from the Laki 1783 eruption. The atmospheric  $SO_4^2$  flux in snow from this eruption over southern and western Greenland was much larger than that of other early 19th century eruptions such as those from 1809 and of Tambora in 1816 (Gao et al., 2007). In SO42- profiles of cores drilled on Penny ice cap (Baffin Island), the Laki 1783 signal is particularly prominent (Goto-Azuma et al., 2002; Zdanowicz et al., 2015). The same is expected to be true on Devon ice cap, as it lies in the same broad latitude band of 65-75°N where maximum SO42- deposition rate were detected in Greenland cores. There are arguments for, and against, each of the SO42- peaks (61.55, 81.55 m) in the DV98.3 core as a potential marker for Laki. If we assign the deeper (81.55 m) peak to 1783, the inferred mean accumulation rate for 1783-1998 is 0.28 m ice a-1 (0.30 m H2O a-1). This agrees closely with the estimated mean accumulation rates for 1963-1998 and 1912-1998 based on radioactivity measurements and the Katmai SO42- peak. If we assign instead the shallower (61.55) m), narrower SO42- peak to 1783, the estimated accumulation rate for 1783-1998 is 0.21 m ice a-1  $(0.23 \text{ m H}_2\text{O} \text{ a}^{-1})$ . This is lower than the figure for 1963-1998 and 1912-1998, but falls within the range of values obtained at various sites across the upper accumulation area of Devon ice cap  $(0.17 \text{ to } 0.25 \text{ m H}_2\text{O a}^{-1}; \text{ Colgan and Sharp, 2008}).$

In the age model that was finally developed, the shallower and sharper of the SO42- peaks (at 61.55 m depth) was ascribed to Laki 1783 (as stated in Kinnard *et al.*, 2006). The nature of the deeper, broader SO42- peak is uncertain: we found no corresponding broad peak in the DV99.1 EC profile within the range of depths at which it should have plausibly been detected. The choice of the 61.55 m EC peak for Laki was partly guided by the fact that resulting age model gave the best overall fit to other, deeper time EC markers in the core, and also resulted in a better agreement of the Holocene  $\delta^{18}$ O profiles between the DV98.1 and DV99.1 cores (see supplement to the revised manuscript), and with other cores previously drilled on Devon ice cap in 1972-73. The confusion in the submitted manuscript arose from the fact that we used an earlier age model iteration in which the Laki 1783 signal had been assigned to the deeper (81.55 m) SO42- peak. It is important to stress, however, that regardless of which of the two SO42- peaks is assigned to Laki 1783, the main features of the DV98.3 SO42- profile remain the same (see figure below), partly because the age model is also constrained by the 1963 radioactive marker. With either time scale,

the profile shows a rise in  $SO_4^{2^-}$  in the mid-20th century, peaking in the 1970 or 1980s, followed by a decline. For the argumentation in the present study, this is the relevant point.

Above: The DV98.3 SO42- profile, averaged over depth intervals corresponding to ~annual increments, and plotted using two different age models. Model (a) assigns the SO42- peak at 61.55 m depth to the Laki 1783, while model (b) assigns the SO42- peak at 81.55 m depth to this same eruption. The sharp SO42- peak ca 1847 in (b) is the one that is assigned to Laki in (a).

The 63.72-m long **DV2000 core** was drilled at the same location (within GPS position error; ± 20 m) as the DV98.3 core. This core was used entirely for the analysis of trace metals such as Pb, and the mechanical decontamination protocol used (described in Zheng et al., 2006) did not leave archive material for other analyses such as major ions or  $\delta^{18}$ O. The DV2000 core was correlated with the DV98.3 on the basis of measurements in the DV2000 that allowed identification of the 1958 (16.5 m depth) and 1963 (13.5 m depth) radioactive layers (Krachler et al., 2005). The DV2000 core was estimated to extend back to 1842. In the detailed comments, the reviewer questions the validity of using the age model of the DV98.3 core for the DV2000 core. This would be a serious issue if our aim was to correlate interannual-scale (or higher-resolution) features in parallel cores, as the signal-to-noise ratio in  $\delta^{18}$ O or glaciochemical signals can be large even with only a few meters of separation (e.g., Fisher et al., 1998). However in this paper we are chiefly concerned with the longer-term (lower-frequency) variations in the ice-core data, which is why we present 10-year averages of the data in Fig. 4 of the manuscript. The reason we discussed the DV2000 Pb record was to emphasize the fact that both SO42- and Pb ice-core records from the summit of Devon ice cap show clear evidence of enhanced anthropogenic aerosol deposition related to fossil fuel combustion emissions in the 20th century, peaking approximately in the 1970 or 1980s. Even if the chronology of the DV2000 was offset relative to that of the DV98.3 core by a few years, it would in no way invalidate this observation.

Regarding the potential effects of summer percolation on the signals in the DV99.1 core:

The long-term (~55-year) mean altitude of the equilibrium line (the ELA) in the summit region of Devon ice cap, based on mass balance measurements going back to the early 1960s, is 1150 m a.s.l.. The highest ELA measured in any given year during this period was 1700 m a.s.l. (D. Burgess, GSC, pers. comm.). Hence for most of the period considered in this study, the DV99.1 site probably lied above the ELA, but within the percolation zone where ice layers can be formed. A detailed core stratigraphic log of the DV99.1 core compiled by R. Koerner indicates that, apart from discrete ice layers and occasional ice glands, most firn layers did not have an icy matrix, i.e. they did not show evidence of being permeated by percolating meltwater. It is therefore likely that the DV99.1 site is, and has been, above the saturation zone for the time period considered here, and also above the superimposed ice zone, which only extends up to ~1400 m a.s.l. presently (Gascon et al., 2013), and probably lower during the colder 19th century. At the summit of Penny ice cap (1930 m a.s.l.), where the firn is of comparable thickness as at the DV99.1 site (> 60 m) but summer melt rates are much higher, all meltwater refreezes in the firn, and there is no net less by runoff (Zdanowicz et al., 2012). So it is very unlikely that there was any net loss of meltwater and impurities at the DV99.1 site in the late 1990s, or over the longer time interval considered in this study. Any summer meltwater produced at this site would have refrozen in situ inside the firn.

Note that the melt record in the DV99.1 core was incorrectly presented the submitted manuscript: the cited melt rates were too large. The original data can be found as an electronic supplement to Fisher *et al.* (2012) *Glob. Plan. Change* 84-85: 3–7. The record consists in data averaged over depth intervals corresponding to ~5-year steps, and extends to 1992. These data have been replotted on the correct scale in the revised manuscript. The estimated 5-year mean volumetric percentage of icy features in the DV99.1 core (the proxy used for summer melt rates) increased from 5 % (median 4 %) between 1783 and 1850, to 21 % (median 20 %) from 1850 to 1992. Within the latter period, there were three 5-year intervals during which the melt feature percentage exceeded 50 %.

The depth at which meltwater could percolate in firn at the DV99.1 site is not known precisely over the time period covered in the rBC record. As mentioned above, the thickness of the firn zone there is >60 m. This is much more than the ~25 m of the firn zone at Lomonosovfonna summit, Svalbard, for example (Kekonen *et al.*, 2005). If we accept the estimated depth range of 0.5-2 m for meltwater-induced ion relocation at Lomonosovfonna summit for 2000-07 reported by Vega *et al.* (2016), then it is highly unlikely than ion relocation could be deeper at the DV99.1 site. The summit of Devon ice cap is >600 m higher than the Lomonosovfonna summit (1250 m a.s.l.), has a much lower mean annual surface temperature (-22°C, compared to ~-10 to -12 °C at Lomonosovfonna; W. van Pelt, *pers. comm.)*, and the 10-m firn temperatures on Devon ice cap were, in 2012, < -15 °C (Bezeau *et al.*, 2013), while those at Lomonosovfonna were within -2 to -3° of zero C in 1997 (van de Waal *et al.*, 2002).

Attempts were also made to quantify post-depositional deposition of ions and/or particles by melt/percolation on Penny ice cap on Baffin Island (66° N; Grumet *et al.*, 1998; Zdanowicz *et al.*, 1998), where estimated summer melt rates over the last 150 years were much higher (40-100 %) than at the DV99.1 site (Zdanowicz *et al.*, 2012). On Penny ice cap during the mid-1990s, ions and particles were estimated to be redistributed over depths of 3-5 m. A plausible, conservative estimate of the maximum melt-induced relocation depth at the DV99.1 site for the time period of interest might therefore be 3 m (firn depth). With a mean accumulation rate of 0.16 m H2O a-1 at

the site, this would imply that impurities could be offset by meltwater percolation in the core by 5-8 years relative to their true depositional depth/age, and probably less for rBC particles given their hydrophobicity. In this paper, we focus on inter-decadal variations in the ice-core parameters (10-averages of the rBC data). With such a wide time-averaging window length, the effects of impurity relocation by meltwater largely even out. Of course, melting-refreezing in the snow/firn may have impacted the rBC record otherwise, by reducing the detection sensitivity of large rBC particles in the melt-affected sections, as discussed in the paper. However we do not find a simple correlation between the rBC concentrations in the DV99.1 core and the melt index (see Fig. 4 in manuscript). Without additional evidence, we can not easily separate the possible effects of melt on the rBC record from those of possible changes (or lack thereof) in aerosol inputs. We therefore try to account for both types of factors in the paper.

As a footnote to the discussion above, decades of experience working with ice cores from Canadian Arctic ice caps have taught us that these records should *mostly, if not only, be analyzed for temporal trends at decadal or longer time scales*, owing to low accumulation rates, or the effects of melt, or both (see Koerner, 1997). Only in the uppermost few tens of meters of some cores from Devon and Ellesmere Island are isotopic or glaciochemical signals sufficiently well-preserved to allow for analysis of these records at interannual time scales (e.g., Kinnard *et al.*, 2006).

Bezeau et al. (2013) J. Glaciol. 59: 181-991. Clark et al. (2007), J. Geophys. Res. 112, D01301, doi:10.1029/2006JD007471. Colgan and Sharp (2008) J. Glaciol. 54, 28-40. Fisher et al. (1998) Science 279(5351): 692-5. Gascon et al. (2013) J. Geophys. Res. 118: 2380-2391. Goto-Azuma and Koerner (2001) J. Geophys. Res. 206(D5): 4959-69. Goto-Azuma et al. (2002) Ann. Glaciol. 35: 29-35. Grumet et al. (1998) Geophys. Res. Lett. 25: 357-360. Kekonen et al. (2002) Ann. Glaciol. 35: 261-265. Kinnard et al. (2006) Ann. Glaciol. 44: 383-389. Koerner (1997) J. Glaciol. 43(143): 90-97. Krachler et al. (2005) J. Environ. Monitor. 7: 1169-1176.m Kudo et al. (1998). J. Environ. Radioactivity 40 (3): 289-298. Pinglot et al. (2003) J. Glaciol. 49: 149-158. van de Waal et al. (2002) Ann. Glaciol. 35: 371-378. Vega et al. (2016) The Cryosphere 10: 961–976. Zdanowicz et al. (1998) Tellus 50B: 506-520. Zdanowicz et al. (2012) J. Geophys. Res. 117, F02006, doi:10.1029/2011JF002248 Zdanowicz et al. (2015) Sci. Tot. Environ. 509-510: 104-114. Zheng et al. (2006) J. Environ. Monitor. 8: 406-413.

**Specific comments, and authors' responses:**

Note: Many of the specific comments below are addressed in our discussion of the age models for the cores, given above. Therefore, where appropriate, we refer the reviewer to this discussion to avoid unnecessary repetition.

Page 2: L. 25: How deep was the core, did you reach bedrock. Is this the same 170.6 m long core as described by (Zheng et al., 2007) as D1999 core?

Yes it is. This information is now in Table 1 of the revised paper.

**Page 3: L. 6: Surprising to see half of the core consumed for EC measurements and low resolution $\delta^{18}$ O analyses. What is the diameter of the cores?**

To clarify: Few analyses were performed on the DV99.1 core *at the time it was drilled* (1999). Coring at the DV99.1 site had been done chiefly as a technical test for a drilling rig. The original core diameter was 9.8 cm. The topmost few meters were of very brittle firn and were not preserved at the time of drilling. Electrical conductivity (EC) measurements were done as a matter of routine using a hand-held device in the field, chiefly with the goal of detecting potential volcanic signatures. Sub-sections of the core wer e cut by band-saw and sent to the Univ. of Copenhagen for  $\delta^{18}$ O analysis. This consumed about 1/3 of the core diameter. Later, in 2007, some archived cores were sampled at irregular depth intervals below 29 m by J. Zheng for trace metal analyses (see Zheng *et al.*, 2007; *Glob. Biogeochem. Cyc.* 21, GB2027, doi:10.1029/2006GB002897), and this consumed >1/2 of some parts of the remaining cores. Sampling for these trace metal analyses was done without an ice-core melter, and involved a lengthy decontamination procedure that used up a considerable volume of firn and ice for these sections, such that what remained of those sampled core sections for rBC/microparticle analysis was very limited.

**L. 9: What is the reference for the initial age estimate?**

See discussion of age models above. The initial age model development for the DV99.1 core is described in Kinnard *et al.* (2006) and Krachler *et al.* (2005), cited in the paper.

L. 10-11: Zheng et al., (2007) report that the core quality was good for the entire core D1999? Is this the same core? If so, which statement is correct? Why don't you add a table with all meta data for ice cores and analyses you discuss in your manuscript?

There was an error in the submitted manuscript on page 3, L8: The part of the DV99.1 archive core that was shipped to Curtin University for rBC analyses reached 48 m depth (not 38 m as was wrongly stated). This depth range was chosen based on the age model for the core to ensure that the rBC record thus produced would extend back to the early to mid-18th century. The uppermost 2.8 m of the DV99.1 core were made of very crumbly firn, and could not be preserved. Below this, the core was recovered in increments of length varying from 0.4 to 1.1 m (average 0.9 m for the part of the core considered in the present study). At depths greater than 48 m, some of the core archive pieces were fractured in many pieces inside the layflat bags.

A table with the metadata on ice cores and analyses was added to the revised manuscript, as suggested.

**L.10: Why should unconsolidated snow not be useable for analyses? It is virtually impossible to contaminate with BC and analyses could have easily been performed with an SP2 on discrete samples.**

The topmost 2.8 m of the core were not preserved at the time of drilling, as they were in poor condition. Hence they were simply not available for analysis. Concerning the physical state of the cores, see responses to previous three comments above. The reviewer's assertion that "it is virtually impossible to contaminate analyses with BC" is incorrect. BC aerosols are everywhere. Because BC has an atmospheric residence time on the order of days, the concentrations found in and on the polar ice caps are of course much lower than at other locations closer to sources. Extreme care is therefore still required for these analyses.

L12: When did the analysis take place? Over how many days, weeks, months? Did you observe sublimation on the ice surface after >15 years of storage?

Analyses were conducted between 6-11 December 2012. Despite being packed in thick, closefitting lay-flat, sublimation/refrost was observed as is to be expected after >15 years of storage. The ice core pieces were therefore carefully cleaned with acid-cleaned ceramic knives and any loose refrozen material was scraped off prior to analysis. The core runs had also broken into several smaller pieces over time which were carefully measured and recorded. The ends of each core piece were also carefully cleaned to avoid contamination by particles. This work was carried out in a class 100 cold room.

**L13: What was class 100? The cold room? The lab space?**

**See below.**

L12-14: I am missing references and I have never heard of an Advanced Ultra-clean Environmental facility. Is this the first time you are performing this kind of analyses in this lab?

The name of the clean room facility changed from the Advanced Ultra-clean Environmental (ACE) facility to the Trace Research Clean Environmental facility (TRACE). A number of publications refer to the space as ACE and the newer publications as TRACE (see below). The facility consists of a large class 100 space containing multiple class 10 laboratory modules including a -20°C walk-in freezer within a general lab space (also class 10). The space was specifically designed for trace metal and particle work on ice cores (e.g., Burn-Nunes et al. 2011, Ellis et al., 2015, 2016; Tuohy et al., 2015; Vallelonga et al., 2017). Ice core preparation was carried out in the walk-in freezer, while processing in the CFA system was conducted in the general lab class 100 space.

Tuohy *et al.* (2015) *J. Geophys. Res. Atmos.*, 120, 10,996–11,011, doi:10.1002/2015JD023293. Burn-Nunes *et al.* (2011) *Cosmochim. Acta.* 75, 1-20, doi:10.1016/j.gca.2010.09.037. Ellis *et al.* (2015) *Atmos. Meas. Tech.*, 8, 9, 3959-3969, doi:0.5194/amt-8-3959-2015. Ellis *et al.* (2016) *Geophys. Res. Lett.*, 43(22),11875-11883, doi:10.1002/2016GL071042 Vallelonga *et al.* (2017) *Clim. Past.*, 13, 171-184, doi:10.5194/cp-13-171-2017.

**L14: At which melt rates did you melt the ice? How do you assure the flowrate is constant? It must be difficult with the frequent change of sold ice lenses (40-60% on average) and soft firn. If the flowrate is not constant, how do you correct for this?**

On each day of analysis, a log journal was created. Every piece of core was carefully measured in its length prior to analysis. During CFA, the time of each break between two ice core pieces was recorded, allowing us to adjust the rBC record of each piece based on the time-depth log. This is the same method that has been used for thousands of meters of ice core analysis in the USA at the Desert Research Institute in Reno Nevada and at other ice core analysis facilities within the USA and internationally. The flow rate to the nebulizer is controlled by oversupplying a <1 mL debubbling vessel with excess water allowing the instrument to maintain a very constant flow rate. Air bubbles from the ice core help to limit dispersion in the tubing up until the point that the ice core water reaches the debubbling vessel next to the BC instrumentation. The system is essential the same as described in McConnell *et al.* (2002) with the exception that the SP2 desolvation system replaces the ICP-MS.

L14-21: Provide a chart with the analytical setup of all instruments. Provide information on calibration (standard material, linearity, stability) and reproducibility of the results.

This has been added in the description of methods, discussion of uncertainties and in the paper's supplement.

L.17-18: These citations are all for a lab in the USA.

See responses to the four previous comments.

L.20: How does the microparticle content connect to your scientific problem? What is the motivation?

We decided to remove the microparticle data from the revised manuscript, and focus solely on the rBC data.

L. 24-25: Why did you not analyze these other aerosol species directly in DV99.1? There should be half of a core minus 2.5 x 2.5 cm of cross section left. This would allow you attribute with more confidence sources to biomass burning and coal burning. Comparisons to other ice cores drilled at different sites only allow you to compare some general trends. According to your age model you have but one common age marker between these ice-core records (the alleged 1783 signal), strong spatial gradients in accumulation caused by wind erosion and/or melting.

See discussion of age models above.

Page 4: L. 4-12: Is any of this age models published? If so, please add the citation and the timescale name for the ice cores, respectively. It appears 2 of 3 citations at the end of this section are based on Agassiz ice cap. I do not find any figure showing the signatures attributed to Laki and Katmai in Kinnard et al. (2006).

**See discussion of age models above.**

L. 4-12: Since your study is critically dependent on the correct identification of the two reference horizons 1783 and 1963, I expect to see all the data that was used for these attributions. It is very plausible that very large acidity in the Arctic were caused by the Laki eruption (Kekonen et al., 2005) but there may also other large acid layers recorded in the Arctic in e.g., 1765, 1815 (Wendl et al., 2015). Equally, the Arctic nuclear fallout signals are in general much broader (1954-1963, (Arienzo et al., 2016)) than described here. How sharp is your signal compared to these other ice cores? Maybe this could tell you something about potential redistribution of impurities caused by melting.

Most of these questions are addressed in our comments about the development of the age models (see above). The radioactive fallout signature in the DV98.3 core does not show strong evidence for post-depositional "smearing" due to melt (Pinglot *et al.*, 2003). However we do not have a corresponding radioactive profile at the DV99.1 coring site where the annual melt percentage is larger, so we can not use such information to quantify the possible effect of melt on ice-core impurities at the DV99.1 site.

We are aware that there are other plausible candidates for some of the  $SO_4^{2-}$  or ECM peaks seen in the DV98.3 and DV99.1 cores. The assignment of specific eruptions/dates to these peaks is of course presumptive in the absence of glass shards (a search done by S. Kuehn, Concord Univ., failed to produce any from these cores). The assignments are usually made in such a way as to provide the greatest degree of coherence across cores and coring sites (for e.g., to synchronize Agassiz ice cap and Greenland cores; Vinther *et al.*, 2008, *J. Geophys. Res.* 113, D08115, doi:10.1029/2007JD009143), and also to produce depth-age models that are as smooth as possible (i.e., that do not, for example, require unrealistic changes in historical accumulation rates). It is the best that can be achieved in the absence of additional data to support these age assignments.

L. 15: Provide these EC measurements for the DV99.1 ice core. How reliable is this peak, if it was recorded in the fractured ice-core sections >38m. Was it reproduced by sulfate analyses?

As discussed above, neither SO42-, nor any other major ions, were analyzed in the DV99.1 core.

L. 17: What accumulation rates to you get between year of drilling, 1963 and 1783 for each of the DV ice cores? Consider providing this information in supplement.

See discussion of age models above.

L. 24-26: Please show these common signatures from DV99.1, DV98.1, DV98.3

There is no core DV98.1. This was a typing error in the text. The  $SO_4^{2-}$  and EC signatures in the DV98.3 and DV99.1 cores are shown in the revised manuscript.

L. 26: What do you mean with adopted? How much depth-age models exist? Where are they published? Are there any isochrones between these age-models and ice cores?

**See discussion of age models above.**

L. 28: How much meters apart were DV98.3 and DV2000 drilled? Is it appropriate to use the same chronology for two different cores given that snow fall and snow conservation on summits are varying on very small spatial scales. I would only adopt a timescale for another ice core if this was supported by a number of isochrones.

See discussion of age models above.

L. 31: I doubt the true effective resolution of the measurements is at a mm scale. You may be recording data at a rate equivalent to mm in depth, but you need to account for the uncertainty in the depth registration and dispersion of the signals through mixing (Bigler et al., 2011).

This is probably correct, and the statement has been modified accordingly in the text. However since we subsequently average discrete measurements over depth intervals representing years to decades, the effect of the uncertainty in the specific depth registration of individual data points is negligible in the interpretation. We are not attempting to interpret features in the rBC record at sub-annual scales.

Page 5: L. 4-12: Add analytical uncertainties as well.

L. 6: (e.g. wind scouring, melt induced relocation). According to Kinnard et al. (2006) average melt rates in D99 are 50% after 1850 AD, which appears to me a significant factor that could modify the impurity records one way or the other.

See discussion of melt-percolation effects above.

L. 9-18: This approach assumes the layer thickness variation is the only source of uncertainty in estimating the age error. It assumes the Laki event is correctly attributed, and must take into account some prior knowledge of the snow accumulation rates and its variability. I don't see how you can estimate the variability "between reference layers of known age" without being able to count annual layers.

As we made clear in the submitted manuscript, when constructing estimates of the uncertainty for time-averaged ice-core parameters (rBC or other), we took into account multiple sources of error, not just the natural variability of the snow accumulation rate and its effect on dating. Dynamically-induced variations in the vertical strain rate over time at the coring site can also result in errors in the estimated ages, but we have no data for quantifying such effects over centuries. The density profiles in the DV98.3 and DV99.1 cores indicate that attenuation (layer-thinning) rates are very nearly linear over the range of depths of interest for this paper, so we have no basis for assuming important variations in vertical strain rate, such as those that might occur, for example, in the accumulation zone of a surging or tidewater-terminating mountain glacier.

The procedure for estimating the uncertainty in the core chronology between reference horizons of known or assumed age does not actually require that annual layers be identified. What we have done do is simulate, by a Monte Carlo procedure, how interannual stochastic variations in snow accumulation (with a plausible frequency spectrum) would cause the age-depth relationship to deviate from that predicted by either a constant accumulation rate model, or some other prescribed age model. By repeating the procedure thousands of times, a population of possible alternative age models is produced, and from these, an envelope of probability for the age of layers between reference horizons is constructed. Although this was not specified in the submitted manuscript (for brevity's sake), we also assigned, in this procedure, possible depth registration errors to the 1963 and/or 1783 age markers, and these errors were taken into account when building envelopes of age probabilities. The probbaility envelopes bracket, for each possible age, the depth interval within which the layer of corresponding age is most likely to be found (or the probable age interval that brackets each given depth). We then used these probabilities as input to a separate Monte Carlo simulation, in which we calculated, for each alternative age model, a resulting time-averaged data series (for example, or rBC). One issue that is not explicitly included in the procedure described above is the possible existence of discontinuities (hiatus) in the stratigraphy of the ice cores due to wind scouring. But since we allow for a broad range of possible variations in the accumulation rate, and we are only interested in the dominant features of the ice-core records at interdecadal scales, this is unlikely to have a large impact on our interpretation.

Note: To make Fig. 4 more easily readable, we removed, in the revised manuscript, the lines that represented the 95 % confidence limits which overcrowded the figure. For the rBC data, these are shown instead in the supplement to the revised manuscript.

Page 6: L. 17-18: Microparticle concentrations do not follow the same trend than nssS and Pb but have a clear step-function. On which observation do you base your attribution of "anthropogenic pollution"?

We removed the microparticle data from the revised manuscript, and focus solely on the rBC data.

L. 22: Such source attributions would strongly benefit of having all parameters analyzed on the same core.

Agreed, which is why we recommended, in our concluding comments, that this be done in a future ice-coring effort.

L. 23: Black carbon concentrations

Text has been changed.

Page 7: L. 12: Necessary not only legitimate

Text has been changed.

Page 8: L. 6-17: Are the sections in the ice with the increased melt layer occurrence believed to be the periods experiencing more melting? Or are they accumulating the meltwater (plus impurities) from the ice sections above? In other words: how deep does percolation go? Is there surface runoff carrying impurities away?

See discussion of melt-percolation effects above.

Given that DV is only 700 km away from NEEM and Humboldt ice cores and all ice cores agreeing on showing strong BC deposition in early 20th century I tend to believe the differences in BC deposition is not from differences in atmospheric burden, but from some aspect specific to the Devon ice cap. The low elevation and observed melt features appear to make a strong case that the impurities along the ice core may be subject to severe loss and/or redistribution in particular during warmer time periods (e.g. Arctic warming 1920-40s (Yamanouchi, 2011), which would smear and bias any atmospheric information.

There is a risk, in making such an argument, of assuming the answer. Our own analysis of air trajectories, presented in the paper, suggests that Devon ice cap is actually affected by a different mixture of aerosol sources than Greenland. In preparing this manuscript, we have tried to avoid making any presumptions about what the DV99.1 rBC record over the past two centuries *should look like*. We did not take it for granted that it should conform to what is observed in Greenland records. It is possible, as the reviewer suggests, that the DV99.1 record differs from the Greenland records due to post-depositional processes, and we acknowledged this possibility in the paper (see also discussion of melt-percolation effects above). But there are also reasons (described in the paper as well) to think that it may differ from Greenland records because of different aerosol source mixture and/or deposition history. Hence we can not simply dismiss the DV99.1 record or rBC deposition available from the Canadian Arctic. Future efforts to duplicate it may prove it to be unreliable, or they may not. We present the DV99.1 rBC record for what it is, with its limitations and caveats, and offer possible competing explanations for why it should contrast with the Greenland records.

L. 18-28: Post-depositional coagulation moving BC sizes out of the detectable range also seems a very plausible explanation for reduced BC recovery during especially warm periods. Low reproducibility of replicate measurements when samples were subject to melting and freezing cycles is reported by several research groups performing BC analyses in ice and snow (apparent "loss" rates in the order of 50%); as a result performing BC analyses on samples that have been refrozen is strongly discouraged (see e.g. (Lim et al., 2014;Wendl et al., 2014)). The ice-core samples used in this study were not subject to refreezing in the sense that they were not melted, then refrozen, prior to analysis. However the core itself contained melt features, which may have reduced the detection sensitivity of rBC particles, as we discussed. If this implies that any core containing substantial melt features is unsuitable for rBC analysis by the SP2 method, then similar rBC records that would be developed from Svalbard ice caps (for example), where annual mean temperatures are much higher than the Canadian High Arctic, should in the future be regarded as even less reliable than the DV99.1 record. But one way to establish if this is the case, or not, is to develop such records and compare them in order to establish their degree of coherence.

Page 9: L. 2-9: This may be true if you compared Devon Ice Cap with Central and Southern Greenland ice cores, but NEEM and Humboldt are just 700-800km away from Devon and are thought to have largely similar source regions for aerosols and precipitation (Zennaro et al., 2014).

**See our response above to comment about page 8, L 6-17.**

Page 10: L. 2-26: How meaningful is such a comparison given the low degrees of freedom (resulting from decadal data), dating uncertainties and the inability to differentiate industrial from BB BC in the Devon ice core? Are these correlation stables if you varied binning, removed the common declining trend over most of the 20th century?

We chose to remove the correlation maps and the corresponding part of the discussion (incl. Fig. 10) from the revised paper, as the correlations were weak, at best.

L. 29-31: This is not surprising; as you outline below: K+, NH4+ and BC have multiple sources and probably also different chemical properties making them more or less susceptible to melt-induced relocation.

The sentence was modified to express this more clearly.

Page 11: L. 2: Could these low numbers for the most recent time period indicate some loss from melting caused by the rapid warming of the Arctic? To my knowledge and supported by Figs. 5 and 6 concentrations of rBC lower than during the pre-industrial baseline are not recorded for any other ice-core in the Arctic.

The situation is not as unique as it seems: In the ACT2 record from southern Greenland (shown in our manuscript), average rBC levels in the first three decades of the 19th century (1800-1830) were lower than in the late 20th century (1960-2000). It is difficult to establish detailed comparisons because only annual averages of the arithmetic mean rBC concentration in Greenland cores have been reported. Comparing geometric mean concentrations would be more revealing. Net losses of water by runoff at the DV99.1 site are unlikely: see discussion of melt-percolation effects above. Also, in the 1960-1990 part of the core where the lowest rBC concentrations were measured, the frequency of ice layers was lower than in deeper core sections with higher rBC concentrations (see revised manuscript).

L. 15: Given the potential limitations from inadequate nebulization, potential loss and redistribution of impurities, I strongly doubt that this value is a realistic approximation of the true atmospheric BC influx.

We stressed in the revised text that our figures are likely lower-bound estimates.

L. 25: Note that you use the same abbreviation EC for both electrical conductivity and elemental carbon.

This has been corrected by avoiding EC to designate elemental carbon.

Page 12: L1-2: Does Humboldt show similar melt features than Devon Ice cap? The agreement between Humboldt and the other Greenland records (in both BC and nssS) seems very high.

This part of the text was removed to shorten and simplify section 4.4. of the manuscript. We are not aware of any published physical stratigraphic record from the Humboldt ice-coring site, or, for that matter, for many other Greenland cores in which rBC has been measured, so we can not comment on this aspect.

L9-12: None of the emission inventories or any ice core suggests that mean BC emissions from 1960-1990 were below preindustrial (i.e. before 1850) levels, as the Devon core seems to imply.

See response to comment about page 11, L2, above.

**Figures:**

Fig. 4: Extend the x-axis to include your only reference marker in 1783 AD. Units in panel b are missing and it is nssSO42-.

This Fig. 4 has been modified considerably to show the extended rBC record back to 1740, and also the 1783 marker in the DV98.3  $SO_4^{2-}$  profile. The total  $SO_4^{2-}$  and  $nssSO_4^{2-}$  profiles in the DV98.3 are virtually identical over the part of the record considered here, so we have chosen to simply show the total  $SO_4^{2-}$  profile.

I do not see any signature related to the largest VEI 6 eruptions of Katmai, Krakatao, Tambora and 1809, but a huge SO4 signal around 1847 AD. Do you have any explanation? How do you know this is not from Laki 1783 or Tambora 1815?

The  $SO_4^{2^-}$  peak at 1847 was the one that was ultimately assigned to Laki, as dicussed above. Unfortunately, we can not have absolute certainty concerning the identification of volcanic EC or  $SO_4^{2^-}$  peaks since we do not have volcanic glass to confirm these assignments. Even if the  $SO_4^{2^-}$  peak assigned to Laki is incorrectly identified and turned out to be from the 1809 or 1816 eruptions, this would have but a minor effect of the main features of the DV98.3  $SO_4^{2^-}$  record (see for e.g., figure in discussion of age models above).

**How do you calculate $K^{+}_{BB}$ ?**

The method that was employed was that of Legrand *et al.* (2016), as was specified in the submitted manuscript (p. 3):  $[K^+]_{bb} = [K^+] - (0.038 \times [Na^+]) - (0.04 \times [Ca^{2+}])$ . Legrand *et al.* (2016) *Clim. Past.* 12, 2033–2059, doi:10.5194/cp-12-2033-2016.

We have chosen not to show the  $NH_4^+$  and  $K_{bb}^+$  profiles on the revised version of Fig. 4. These are, however, included these plots in the supplement.

Fig.4 and Fig.6: DV98.3 nssSO4 (panel b, Fig 4) appears to be different from nssS (Fig. 6), the latter is peaking in the 1960-1980s, the first starts to peak only after 1980. Which one is correct?

The discrepancy was due to a plotting error in one of the figures, and has been corrected.

Fig. S3: Check the lower panel. There should be only one y-value for a given depth value

Corrected in revised paper.

**1** Historical black carbon deposition in the Canadian High Arctic:**

**2 A >250-year long ice-core record from Devon Island**

| 3
4 | Christian M. Zdanowicz 1 , Bernadette C. Proemse 2 , Ross Edwards 3,4 , Wang Feiteng 5 , Chad M. Hogan 2 , Christophe Kinnard 6 and David Fisher 7 . |  |  |  |  |  |
|--------|-----------------------------------------------------------------------------------------------------------------------------------------------------------------------------------------------------------------------------------|--|--|--|--|--|
| 5      | 1 Department of Farth Sciences, Unpsala University, Uppsala, 75646, Sweden                                                                                                                                             |  |  |  |  |  |
| c
c | 2 Sehool of Piological Sciences, University of Teamonia, Hohart, TAS7001, Australia                                                                                                                                    |  |  |  |  |  |

[revised manuscript text omitted]

- do

---

## Referee Report (RR2)

General comments:

After the first revision the paper has been significantly improved in terms of clearness and a more honest interpretation of the still debatable results was given. Particularly, despite the evident limitation in the depth-age model for the DV99.1 core, now it seems more reasonable. The decadal interpretation of the results seems the right choice because of the lack of annual resolution and the uncertainties in the depth-age model.

However, there are still some points that require more precise explanations and, particularly the SP2 calibration procedure and data analyses description, still need some efforts and clarifications.

In the following some comments:

- Figure 2: Please add the Katmai EC marker in the DV99.1 age plot (even if uncertain). The long term
- It was not possible to analyze ionic species: Why? During the melting no discrete samples were taken? The instruments were not available?
- The only EC measurement available is the one done with the hand held device? Was the core in good conditions below the 38 m? How accurate are those measurement?
- Figure S10: there is a cross in the period 1840-1850, is it an outlier? If yes how was it defined?
- Figure S13: rBC and elemental carbon concentrations needs correction factors in order to be directly compared. In the caption and in the y-axis you wrote «rBC or Elemental carbon», please report specifically for which Greenland ice core was measured rBC and/or elemental carbon (if some of them were analyzed not with an SP2 you shouldn't call the resulting concentration with «rBC»). Therefore, the same correction has to be done in Figure 5 of the main text.
- Line 46/47: please consider adding the utilization of the CEATC nebulizer as a possible source of losses on real BC particles during the analyses.
- Line 81: add the word «concentration» after «rBC»
- Line 100: please add that the crumble firn part was not preserved at the time of the drilling.
- Line 124: what does it mean was «essentially» the same? Did all of them use an SP2? Did they use different nebulizers?
- Line 186/187 and Figure S7: it is known that melting and refreezing cycles tend to reduce the Delta18O oscillations in snow/ice; so how can you confidently speculate that 40-45% of snow was annually removed considering that both wind and melting affects the DV99.1 site?
- Change the scale of the Figure S6b in order to make the D18O oscillations more evident (the same in Figure S7).
- In order to understand if the CETAC system contributed in hiding the increase of the rBC concentration in the last century due to the increase of the particles dimensions caused by coagulation during melting and refreezing cycles, it would be very important (and the results shown in this paper absolutely need this) to compare the mass size distribution of BC particles in

the real sample in a period with very low melting features and in one with very high volumetric percentage of icy melt features.

- o Line 268: In the decade 1780-1990? Please correct these dates.
- o Line 289: I would rather use more general terms saying «wet deposition of BC containing particles» (not only the hydrophilic part of them).
- o Line 470: underestimation of 20-40% without considering the 25% of efficiency of the CETAC system?

**SP2 Calibration**
- o How did you analyze the data? Did you write your own code for analyzing the SP2 raw data? Or did you use any prepared toolkit?
- o You used the «Ebony MIS, EB6-4 K» material for SP2 calibration, what is that? Is there any reference paper about it? How the particles mass was measured? Cite the work that has characterized it, e.g. for its «effective density» (as Gysel Martin did for Aquadag and Fullerene, the most used calibration materials, in «Effective density of Aquadag and fullerene soot black carbon reference materials used for SP2 calibration"). Moreover, using the CETAC system with its typical cutoff in the nebulization efficiency it is important that the peak of the mass size distribution of the calibration material is in the window of the highest nebulization efficiency, otherwise you cannot trust the absolute concentration values.
- o Please be more precise in explaining the calibration procedure of the SP2. What is the «Response»? Did you get the SP2 internally calibrated from DMT and well aligned before using it?
- o Did you perform the SP2 alignment and the laser's beam shape check prior to the analyses? Was the laser beam into the TEMoo mode as reported by the instrument manual?
- o You said that «the CETAC efficiency was typically 25%». What does it mean? The nebulization efficiency? Does it mean that only the 25% of the sample was nebulized and carried in the SP2? Therefore the measured rBC concentration could be 25% of the actual value?
- o Could you provide the number concentration to mass concentration ratio (from the SP2) for the standards and for a part of the real sample? This is very important if you are not referring to a paper describing in details the new calibration material that you used (in terms of size distribution measured for instance with an SMPS).
- o Why do you call the calibration «External»?
- o Do you sonicate the standard solutions before each calibration? Do you keep a standard solution with a very high concentration and then dilute it every day for making the low concentration ones? And how do you know that the CETAC system was not the responsible for the decrease in the calibration slope? Were the most recent sections of the core analyzed before or after the decrease in sensitivity?

**Typing errors:**
- o Please refer to «rBC» only when reporting the mass concentration measurements done with the SP2 (e.g. remove the «r» before «rBC particles» in line 300, before showing the TEM image in the supplementary material, in the caption of the Figure S2…). Be more specific in the caption of

the Figure S2: explain what the aggregates shown are, maybe graphite or soot. Insert a space before (ppb) in the caption of this figure.

- Write the dates in a coherent way in the figures «S3».
- Please add the «melting speed» in the supplementary material.
- Page 7, line 199: remove the comma.
- Line 142 spelling error: «potemtial»

---

## Author Response (AR2)

**Specific comments, and authors' responses:**

*Figure 2: Please add the Katmai EC marker in the DV99.1 age plot (even if uncertain).*
Added as suggested.

*It was not possible to analyze ionic species: Why? During the melting no discrete samples were taken? The instruments were not available*
At the time the instruments and funding were not available to measure ionic species.

*The only EC measurement available is the one done with the hand held device?*
Yes.

*Was the core in good conditions below the 38 m?*
See our earlier response to the first round of revisions: There was an error in the original version of the manuscript. The part of the DV99.1 archive core that was shipped to Curtin University for rBC analyses reached 48 m depth (not 38 m as was wrongly stated). This depth range was chosen based on the age model for the core to ensure that the rBC record thus produced would extend back into the early 19th to mid-18th century. The uppermost 2.8 m of the DV99.1 core were made of very crumbly firn, and could not be preserved. Below this, the core was recovered in increments of length varying from 0.4 to 1.1 m (average 0.9 m for the part of the core considered in the present study). At depths greater than 48 m, some, but not all, of the core archive pieces were fractured in many pieces inside the layflat bags.

*How accurate are those measurements ?*
The hand-held EC measurement system that was used is an earlier version of the unit designed and sold by Icefield Instruments Inc. (see: http://www.icefield.yk.ca/portable-ecm-unit.html). When we used this system in the field, we started by manually scrapping off a few mm of ice from the core sections to be measured using a stainless steel knife, then we ran the electrodes along the length of each core section at a speed of approximately 2 cm per second, which, based on the 19 Hz digitizing rate of the instrument, gives approximately one measurement of EC per mm along the core surface. We did not attempt to calibrate the voltage readout because we were only interested in changes in relative EC along the core, so we can not quantify the accuracy in mV. There are several possible sources of uncertainty in such EC measurements. One is the heterogeneity of $H^+$ distribution in the ice, which may result in a variable EC on different faces of the same core section (this is true even if a different EC measurement system is used). If there are breaks on the core surface, or the surface is very uneven (as can be the case with crumby firn), the contact of the electrodes with the firn or ice may be poor. This is why we only started the measurements at 12.5 m depth. The EC recordings from each segment were later spliced together to produce a continuous profile, registered against the core depths. To get an idea of how the EC profile of the top 50 m of the DV99.1 core compares to deeper sections, see figure on next page.

[Figure]

Above: Comparison of the EC profile in the DV99.1 core down to 100 m depth (left) with the top 50 m section enlarged and presented in the manuscript (Fig. 3c). The image at right is the raw EC data. At left (panel from Fig. 3), the ECM data were smoothed with a running mean equivalent to a 1-cm vertical resolution, in order to make the main features of the EC profile appear more clearly, which explains why the range of values on the *x*-axis of these two panels are different.

*Figure S10: there is a cross in the period 1840-1850, is it an outlier? If yes how was it defined?*

The whiskers on each box-plot span approximately 99 % of the probability distribution of rBC concentrations in each decade, and values that lie outside of the probability range are classified as outliers. There were rBC outliers in all the 10-year periods. When we prepared Fig. S10 we chose not to display them because they visually cluttered the plot, and what we mostly wanted to emphasize here was how the median and interquartile ranges vary from decade to decade. This particular outlier was accidentally left in the plot. This has now been fixed in the revised version of the figure, and more details added in the caption to specify how the whiskers are defined.

*Figure S13: rBC and elemental carbon concentrations needs correction factors in order to be directly compared. In the caption and in the y-axis you wrote «rBC or Elemental carbon», please report specifically for which Greenland ice core was measured rBC and/or elemental carbon (if some of them were analyzed not with an SP2 you shouldn't call the resulting concentration with «rBC»). Therefore, the same correction has to be done in Figure 5 of the main text.*

We modified the figure to display snow and rBC accumulations rates on Greenland and Devon ice cap as a function of latitude, instead of showing rBC concentrations vs. snow accumulation rates, as before. We excluded the data from Holtedahlfonna core (Ruppel *et al.*, 2014) from this plot, such that only data based on rBC measurements by the SP2 method are now shown. The data shown on Fig. 5 in the manuscript are also only rBC measurements by the SP2 method. We also re-wrote the last paragraph of section 4.4. in the manuscript (L491-L517), to reflect these changes.

*Line 46/47: please consider adding the utilization of the CETAC nebulizer as a possible source of losses on real BC particles during the analyses.*

The sentence is about factors that might explain the observed differences between the DV99.1 rBC record and Greenland records developed by the same method (SP2). Since the Greenland records were also developed using the CETAC system, what really matters is how the effects of melt would reduce the detection efficiency of rBC in the DV99.1 core, relative to Greenland cores with little or no melt. The sentence already makes that point and does not need further editing.

*Line 81: add the word «concentration» after «rBC»*

Added as suggested.

*Line 100: please add that the crumble firn part was not preserved at the time of the drilling.*

Added as suggested.

*Line 124: what does it mean was «essentially» the same? Did all of them use an SP2? Did they use different nebulizers?*

The word "essentially" was removed. The methods were the same.

*Line 186/187 and Figure S7: it is known that melting and refreezing cycles tend to reduce the Delta18O oscillations in snow/ice; so how can you confidently speculate that 40-45% of snow was annually removed considering that both wind and melting affects the DV99.1 site?*

The estimate is based on the comparison of the amplitude of the $\delta^{18}O$ signals between the D98.3 and the DV99.1 cores. The two coring sites are situated at elevations that only differ by 30 m (see Table 1) and it is almost certain that they experience the same range of seasonal temperatures, and therefore the same amount of summer surface melt (but not necessarily the same volume of refrozen meltwater in the firn). In that case the observed difference in the $\delta^{18}O$ amplitude must be accounted for by other factors, such as the rate of net snow accumulation (which also affects the volumetric percentage of refrozen meltwater in firn). The two coring sites are also close to each other (see Fig. 1) and it is very unlikely that precipitation rates would be vastly different over such a short distance. Therefore the most likely explanation for the $\delta^{18}O$ signal differences is wind scouring of winter snow, and this is coherent with the truncation of the most negative $\delta^{18}O$ values in the DV99.1 core compared to the DV98.3 core, as seen before at other sites in the Canadian High Arctic (see references in manuscript).

*Change the scale of the Figure S6b in order to make the D18O oscillations more evident (the same in Figure S7).*
Changed as suggested.

*In order to understand if the CETAC system contributed in hiding the increase of the rBC concentration in the last century due to the increase of the particles dimensions caused by coagulation during melting and refreezing cycles, it would be very important (and the results shown in this paper absolutely need this) to compare the mass size distribution of BC particles in the real sample in a period with very low melting features and in one with very high volumetric percentage of icy melt features.*

Changes in the rBC particle mass distribution and aggregation is a concern. However, we found no evidence of this from the rBC mass probability distribution. In support of this conclusion, we have added a figure to the supplement (Fig. S3), reproduced below. This figure shows the normalized probability distributions of rBC mass in different ice-core sections, some of which date back to the interval ~1803-1814 during which the ice-core melt feature percentage was only 1 % (vol.), and the other spanning the interval ~1943-1963 during which melt feature percentages were as high as 53 % (min. 9 % vol.). As can be seen, the rBC mass have similar probability distributions in all these core sections, and lack any evidence for marked differences that could be attributed to rBC particle aggregation during periods of higher surface melt. The text in section 4.2, L376-L384 of the manuscript, was also amended to account for these results.

[Figure]

Above: Figure S3 from supplement, showing normalized probability distributions of rBC mass determined by SP2 in sections of the DV99.1 core spanning depths 10-12 m, 12-13m, 15-16 m, and 37-38m.  The 10-16 m sections are from years ~1943-1963, while the 37-38 m section is from years ~1803-1814.

*Line 268: In the decade 1780-1990? Please correct these dates.*

Corrected: 1780-1790.

*Line 289: I would rather use more general terms saying «wet deposition of BC containing particles» (not only the hydrophilic part of them).*

Changed as suggested.

*Line 470: underestimation of 20-40% without considering the 25% of efficiency of the CETAC system?*

This was a mistake, now corrected in the text (L477-L478). The possible underestimation of the net rBC deposition rate was evaluated based on the assumption that the rBC concentrations in snow may be underestimated by 60-80 %, (not 20-40 % as we wrote) due to the combined effects of wind scouring and the CETAC system detection inefficiency. To be clear: What we did is to recalculate the net deposition rate of rBC in snow that would result if the true rBC concentrations in snow were 60 to 80 % greater than what was actually measured, and obtained corrected deposition rates of 0.19 to 0.23 mg m$^{-2}$ a$^{-1}$, averaging 0.2 mg m$^{-2}$ a$^{-1}$ (the rounded figure cited in the text). The estimates bracket a possible range of conservative scenarios. For example, if wind scouring removes 45 % of the rBC, and the remainder is also underestimated by 25 % due to the CETAC system limitations, then the true rBC concentration would be underestimated by 65 %, i.e. between 60-80 %.

**SP2 Calibration**

*How did you analyze the data? Did you write your own code for analyzing the SP2 raw data? Or did you use any prepared toolkit?*

The raw SP2 binary files (.sp2) were converted into ASCII files using the Performance Application Programming Interface toolkit (WaveMetrics IGOR) supplied by the SP2 manufacturer (Droplet Measurement Technologies). We used a custom code written in R and CRAN R open-source software to filter broadband incandescence peaks and convert incandescence peaks to mass using the calibration relationships (supplied by Droplet Measurement Technologies). Peak filter parameters recommended by DMT.

*You used the «Ebony MIS, EB-6K» material for SP2 calibration, what is that? Is there any reference paper about it? How the particles mass was measured? Cite the work that has characterized it, e.g. for its «effective density» (as Gysel Martin did for Aquadag and Fullerene, the most used calibration materials, in «Effective density of Aquadag and fullerene soot black carbon reference materials used for SP2 calibration"). Moreover, using the CETAC system with its typical cutoff in the nebulization efficiency it is important that the peak of the mass size distribution of the calibration material is in the window of the highest nebulization efficiency, otherwise you cannot trust the absolute concentration values.*

The EB-6K is a commercially available, 100% carbon black pigment. This material was used in preference to the standards used by McConnell et al. (2007) because of its larger particle size, which is closer to the geometric mean volume equivalent diameter (~200 nm) of BC found in the remote atmosphere. We have added a plot of the probability distribution of particle mass (Figure S3) to the supplement. The rBC particles were within the CETAC aerosol cutoff window. The actual size of the standard particles were also confirmed by transmission electron microscopy (TEM). TEM is really the only way to know the actual size and morphology of the particles.

*Please be more precise in explaining the calibration procedure of the SP2. What is the «Response»? Did you get the SP2 internally calibrated from DMT and well aligned before using it?*

We have changed the *y* axis label in Figure S4 from "response" to "rBC mass $(10^{-15}$g sec$^{-1})$". The SP2 was calibrated and aligned by DMT.

*Did you perform the SP2 alignment and the laser's beam shape check prior to the analyses? Was the laser beam into the TEMoo mode as reported by the instrument manual?*

The laser alignment / laser beam (TEMoo) shape was checked before use with a DMT-supplied beam scan camera and software, following the instruction manuals (DOC-0175, DOC-0229) provided by Droplet Measurement Technologies.

*You said that «the CETAC efficiency was typically 25%». What does it mean? The nebulization efficiency? Does it mean that only the 25% of the sample was nebulized and carried in the SP2? Therefore the measured rBC concentration could be 25% of the actual value?*

The CETAC efficiency is defined as the mass conversion of water to aerosol at the nebulizer transducer section of the nebulizer. The nebulizer efficiency is calculated as the mass flow rate of water entering the nebulizer to the mass flow rate of water exiting the nebulizer drain below the transducer. An efficiency of 25% means that only 25% of the standard or sample mass was carried into the remaining sections of the nebulizer desolvation system. External standards are used to construct a linear relationship between rBC entering the nebulizer and the SP2 determined rBC mass per cm of core melted, or corresponding time period. The external calibration is used to correct for mass transport losses in the nebulizer. This is a standard practice for many instrumental methods. For example ICP-MS, ICP-OES, GC-MS. Nearly all analytical instruments used with an introduction system rely on external or internal (or both) standards to correct for losses and bias in the introduction system.

*Could you provide the number concentration to mass concentration ratio (from the SP2) for the standards and for a part of the real sample? This is very important if you are not referring to a paper describing in details the new calibration material that you used (in terms of size distribution measured for instance with an SMPS).*

We have added a normalized probability distribution plot of particle masses in a standard and an ice core sample to the supplement (now Figure S3) and some additional text on page 3 of the document.

*Why do you call the calibration «External»?*

Good question. "External calibration" is a common quantitative chemistry term. An external calibration is constructed by relating instrumental responses to external standard concentrations etc. External standards are not mixed with the sample itself, hence the word "external" as opposed to "internal". Internal standards are mixed with the sample itself.

*Do you sonicate the standard solutions before each calibration? Do you keep a standard solution with a very high concentration and then dilute it every day for making the low concentration ones? And how do you know that the CETAC system was not the responsible for the decrease in the calibration slope? Were the most recent sections of the core analyzed before or after the decrease in sensitivity?*

No, we sonicated the (high concentration) primary standard bottle before opening it. The other diluted standards were prepared from the primary bottle and well shaken. We are not sure of the cause in the calibration shift, only that it was systematic. However we found no shift in the rBC concentration, as described on page 5 of the supplement. To clarify the position of the shift we have added a high depth resolution figure of the rBC concentration and depths for the different dates and the position of the calibration shift (Fig. S5, also shown below)

[Figure]

Above: Figure S5 from supplement, showing rBC concentrations in the DV99.1 core versus depth. Different colors highlight the ice-core analysis date. The dashed vertical line at ~37 m marks the shift in calibration slope.

**Typing errors:**

*Please refer to «rBC» only when reporting the mass concentration measurements done with the SP2 (e.g. remove the «r» before «rBC particles» in line 300, before showing the TEM image in the supplementary material, in the caption of the Figure S2…).*
Corrected throughout the text, where necessary.

*Be more specific in the caption of the Figure S2: explain what the aggregates shown are, maybe graphite or soot.*
Corrected: "Transmission Electron Microscope image of colloidal carbon black pigment particles (MIS EB-6K) used as standards for calibration of the SP2."

*Figure S3: Insert a space before (ppb) in the caption of this figure.*
Corrected.

*Write the dates in a coherent way in the figures «S3».*
Corrected.

*Please add the «melting speed» in the supplementary material.*
The melting speed (3 cm min$^{-1}$) has been added to the supplementary document, page 2.

*Page 7, line 199: remove the comma.*
Corrected.

*Line 142 spelling error: «potemtial»*
Corrected.

---

## Author Response (AR3)

**Authors' response to referee #3's comments**

**Specific comments, and authors' responses:**

*1) In the paper it is still worryingly unclear the SP2 calibration procedure, which is essential in order to have realistic absolute values of rBC concentration. In your response it is written that the SP2 calibration was performed by DMT and that "We used a custom code written in R and CRAN R open-source software to filter broadband incandescence peaks and convert incandescence peaks to mass using the calibration relationships (supplied by Droplet Measurement Technologies)". So the question is:*

*1a) The DMT supplied the calibration coefficients (as it happens usually)? If yes which calibration material was used by the DMT during the SP2 calibration? If DMT supplied the calibration curve, how the calibration results reported in Figure S4 were used? Only for calculate the nebulization efficiency of the CETAC system?*

There are two calibrations. The first calibration was performed by DMT used DMA mobility size selected Aquadag particles to estimate the single rBC particle mass in relation to broad-band incandescent light intensity. There has been some discussion regarding the variability of incandescent emissivity between different rBC materials – in particular Aquadag versus "as produced" fullerene soot (from Sigma-Aldrich). For the ice core research we have continued to use Aquadag for the broad-band detector relationship to rBC mass in order to be consistent with past ice core analyses. We acknowledge the differences between rBC mass as determined using Aquadag and fullerene soot as discussed by Moteki and Kondo (2010; *Aerosol Sci. Tech.* 44: 663–675). However in this case fullerene soot was determined to be a good standard with regards to the analysis of ambient rBC in urban Tokyo. We do not know whether it would be a good standard for rBC incorporated into the remote the polar ice caps. In many respects these regions are the antithesis of urban Tokyo or any urban atmosphere.

The second calibration, (performed, before, during and after the analysis) was used to determined the relationship between liquid rBC concentration and the aggregate response of the entire system over a constant time step. This is not merely a calibration of the nebulizer, but rather the entire system, including mass transport through the nebulizer and all the sp2 parameters including detector efficiency, gas flows etc. As with many other instrumental platforms which are calibrated against liquid concentration the assumption is that the standard and its matrix is a good approximation of the sample. This is one of the reasons that we tried to choose a standard with a similar mass distribution to the sample. The nebulizer efficiency of the CETAC nebulizer was determined as a day to day check of the nebulizer system stability and is not necessary for the calibration.

We have changed lines on page 3 of the supplemental to read:

"The nebulizer efficiency (~25%, defined as the percent mass of water aerosol exiting the nebulizer chamber into the heated section) was monitored gravimetrically to check the nebulizer stability."

*1b) Moreover, how did you make the standard? Did you assume a 100% content of BC particles in the original standard? This is a key step in order to know the actual BC concentration in a solution. No EC (Elemental carbon) measurement was performed to ensure this (as done for the Aquadag and Fullerene in other studies)? Even for Aquadag the total content of refractory BC particle is not 100%. If you have used the "new" standard for making the calibration of the SP2 then it will be important to make and estimate of the possible degree of underestimation/overestimation of the absolute concentration values caused by the use of a different standard.*

To prepare the standard we first determined the dry mass and refractory rBC by thermogravimetry. Thermogravimetry was chosen over the NIOSH thermal/optical EC analysis since it is typically referenced to sucrose and is not considered a primary analysis. With regards to the thermogravimetric analysis we determined that the liquid standard contains 7% refractory mass, here defined as the stable mass at 400 °C in air. We determined the rBC mass contribution to be 92.7% by subtracting the remaining mass after autoignition at temperatures above 450 °C. After determining the mass at 400 °C the temperature was increased to 480 °C for 40 minutes and then 580 °C for 40 minutes. As the temperature increased from 400 to 480 °C the bulk of the material passed through its autoignition and became incandescent. No visible emissions could be seen. We assume that the mass lost during autoignition was rBC. The mass remaining after 40 min at 580 °C was 7.3% of the refractory mass. Based on the results of the thermogravimetry we prepared the concentration standards by serial dilution in ultrapure water.

We have added further details on page 2 of the Supplement:

"Thermogravimetric analysis of the standard revealed the dry mass to be 7.06% of the liquid mass. The dry mass was stable in air from 200 to 400°C. We have recently analysed a fresh batch of the same pigment and found a comparable dry mass of 7.23% at 400 °C (for 40 min in air). After increasing the temperature to 480 °C (for 40 min) the material glowed red with no visible emissions suggesting an autoignition temperature between 400 and 480 °C. The temperature was increased to a final temperature of 580 °C for 40 min. We found no mass change between the 480 and 580°C stages. The remaining mass after the 580 °C stage was 7.3% of the dry mass. Based on the thermogravimetric analysis we concluded that 92.7% of the dry mass at 400 °C was rBC. Concentration standards were prepared by serial dilution of the pigment with ultrapure water based on the results of the thermogravimetric analysis".

*1c) In the paper by Moteki and Kondo (2010), a comparison of SP2's responses to different calibration material is given. Please use their results to establish a plausible degree of variability of your absolute concentration values if you were to use other standards (just use their slopes variability and change your calibration curve accordingly).*

We do not think that we can simply changing the slopes to match that of Moteki and Kondo (2010) as described by the reviewer. The correct approach would be to determine the optical and morphological properties of rBC in Greenland snow after it has been melted and then make a direct comparison with the Aquadag standard used by DMT. However, our guess is that if the Eboni standard was at the extreme end of the values from the single particle masses our results using the

standard could vary from fullerene soot by a factor of two.  We do not believe this is the case however. Even if this were the case the trend in the time series would not change.

*2) Please show the plot related to the change in sensitivity also using the aging curve, so that it will be much easier to understand for the reader where there could possibly be a higher degree of uncertainty in the results.*

Response: We have exchanged the depth scale for the age scale in Fig. S6.

*3) Why are you using the units "rBC (10...)" in the y-axis of the Fig. S4? Are you sure it is not "Incandescence Signal"? Please add the unit also in the plot of the calibration curve.*

The units are $10^{-15}$ g rBC per sec. Or in this case $10^{-15}$ g rBC per 2 sccm (standard cubic cm per minute). The calibration *y*-axis units are all the same.

We have added the following to the caprion of Fig. S4: "The y axis units for all panels are femtogram rBC per sec which is equivalent to units of femtogram rBC per 2 sccm of air."

*4) You should report the plot of the mass distribution in "dN/dlnDp", not in "dN/lnDp" (use the logarithm of the diameters in the ratios).*

This was a typographical error. We have changed the label to"dN/dlnDp"